# Ephrin A1 functions as a ligand of EGFR to promote EMT and metastasis in gastric cancer

Shuang Li [ID][1,2,3,4,7], Meng Sun [ID][1,2,3,4,7], Yun Cui[1,3,4,7], Dongyang Guo[1,3,4,7], Feng Yang[2], Qiang Sun[5], Yinuo Ding[1,3], Mengjie Li[1,3], Yiman Liu[1,4], Guangshuo Ou [ID][6], Wei Zhuo [ID][1,3,4✉] & Tianhua Zhou [ID][1,3,4✉]

## Abstract

**Distant metastasis is the major cause of gastric cancer mortality, and epidermal growth factor receptor (EGFR) activation plays critical roles in gastric cancer dissemination. However, EGFR targeting therapies in gastric cancer show only marginal effects, and the molecular mechanisms of oncogenic EGFR signaling remain poorly defined. Here, we report Ephrin A1 as a novel ligand of EGFR in gastric cancer. Ephrin A1 facilitates colonization and metastasis of gastric cancer cells in vitro and in vivo via inducing epithelial-mesenchymal transition (EMT). Ephrin A1 directly interacts with EGFR and induces EGFR dimerization, phosphorylation and activation of downstream signaling. Ephrin A1-induced EMT can be rescued by EGFR signaling inhibitors or knockout of *EGFR*, but not depletion of its classical receptor *EphA2*. Moreover, Ephrin A1 protein level correlates with EGFR phosphorylation levels in gastric cancer patients. Collectively, our work uncovers Ephrin A1 as a functional ligand of EGFR and highlights the potential role of the Ephrin A1/EGFR/EMT regulatory axis in cancer metastasis.**

**Keywords** Gastric Cancer; EGFR; Ephrin A1; EMT; Metastasis
**Subject Categories** Cancer; Cell Adhesion, Polarity & Cytoskeleton; Signal Transduction

## Introduction

Gastric cancer is the fifth most commonly diagnosed cancer and the fourth leading cause of cancer deaths worldwide (Sung et al, 2021). Approximately 40% of gastric cancer patients present with metastases, characterized with a 5-year survival rate about 5% (Bernards et al, 2013; Zhang et al, 2020). Therefore, it is urgent to learn more about the mechanisms of gastric cancer metastasis and find effective therapy targets for metastatic gastric cancer.

Epidermal growth factor receptor (EGFR) is an important target for anti-cancer therapies. Aberrant activation of EGFR signaling is associated with many types of cancers (Sigismund et al, 2018; Zandi et al, 2007). Notably, EGFR tyrosine kinase inhibitors (TKIs) against EGFR activation have been approved to treat certain kinds of cancers, such as lung and pancreatic cancers (Lee et al, 2014; Wu and Shih, 2018). Although EGFR is upregulated in gastric cancer, EGFR-targeted therapies have shown marginal clinical effect in gastric cancer patients without any predictive biomarkers (Lordick et al, 2013; Waddell et al, 2013). There may be other regulatory mechanisms for EGFR in gastric cancer. Generally, EGFR is activated by binding of different ligands and leading to signaling cascades, which further regulates proliferation, cell invasion and other oncogenic responses (Appert-Collin et al, 2015; Linggi and Carpenter, 2006). Beside the well-known ligands such as EGF, TGF-α, and amphiregulin, new ligands including angiogenin and connective tissue growth factor (CCN2) have been reported to activate EGFR signaling and may provide new insights for EGFR-targeted therapy (Rayego-Mateos et al, 2013; Wang et al, 2018).

Ephrin family are ligands of Eph receptors, the largest sub-family of receptor tyrosine kinases (Bartley et al, 1994; Holzman et al, 1990). Multiple studies have shown that Ephrin A1 was abnormally expressed in cancers (Beauchamp and Debinski, 2012; Hao and Li, 2020; McCarron et al, 2010). However, the expression and function of Ephrin A1 in cancers are complex owing to cell-type dependent (Beauchamp and Debinski, 2012; Iida et al, 2005; Liu et al, 2007; Yamamoto et al, 2013; Youngblood et al, 2016). Our previous study revealed that Ephrin A1 was upregulated in gastric cancer and involved in metastasis (Zhuo et al, 2019). However, the exact mechanism of Ephrin A1 in gastric cancer metastasis remains largely unknown.

In this study, we identified Ephrin A1 as a new ligand for EGFR. Ephrin A1 binds to and activates EGFR to induce epithelial-mesenchymal transition (EMT) and promote metastasis of gastric cancer cells. The Ephrin A1/EGFR/EMT regulating axis may provide new strategies for targeting therapy in gastric cancer metastasis.

## Results

### Ephrin A1 promotes EMT of gastric cancer cells

To explore the mechanism of Ephrin A1 in promoting gastric cancer metastasis, we detected the expression of Ephrin A1 in

[1]Department of Colorectal Surgery and Oncology and Department of Cell Biology, Center for Medical Research and Innovation in Digestive System Tumors, Ministry of Education, the Second Affiliated Hospital of Zhejiang University School of Medicine, Hangzhou, China. [2]Binjiang Institute of Zhejiang University, Hangzhou, China. [3]Cancer Center of Zhejiang University, Hangzhou, China. [4]Institute of Gastroenterology, Zhejiang University School of Medicine, Hangzhou, China. [5]International Institutes of Medicine, The Fourth Affiliated Hospital of Zhejiang University School of Medicine, Yiwu, Zhejiang, China. [6]School of Life Sciences, Tsinghua University, Beijing, China. [7]These authors contributed equally: Shuang Li, Meng Sun, Yun Cui, Dongyang Guo. ✉E-mail: wzhuo@zju.edu.cn; tzhou@zju.edu.cn

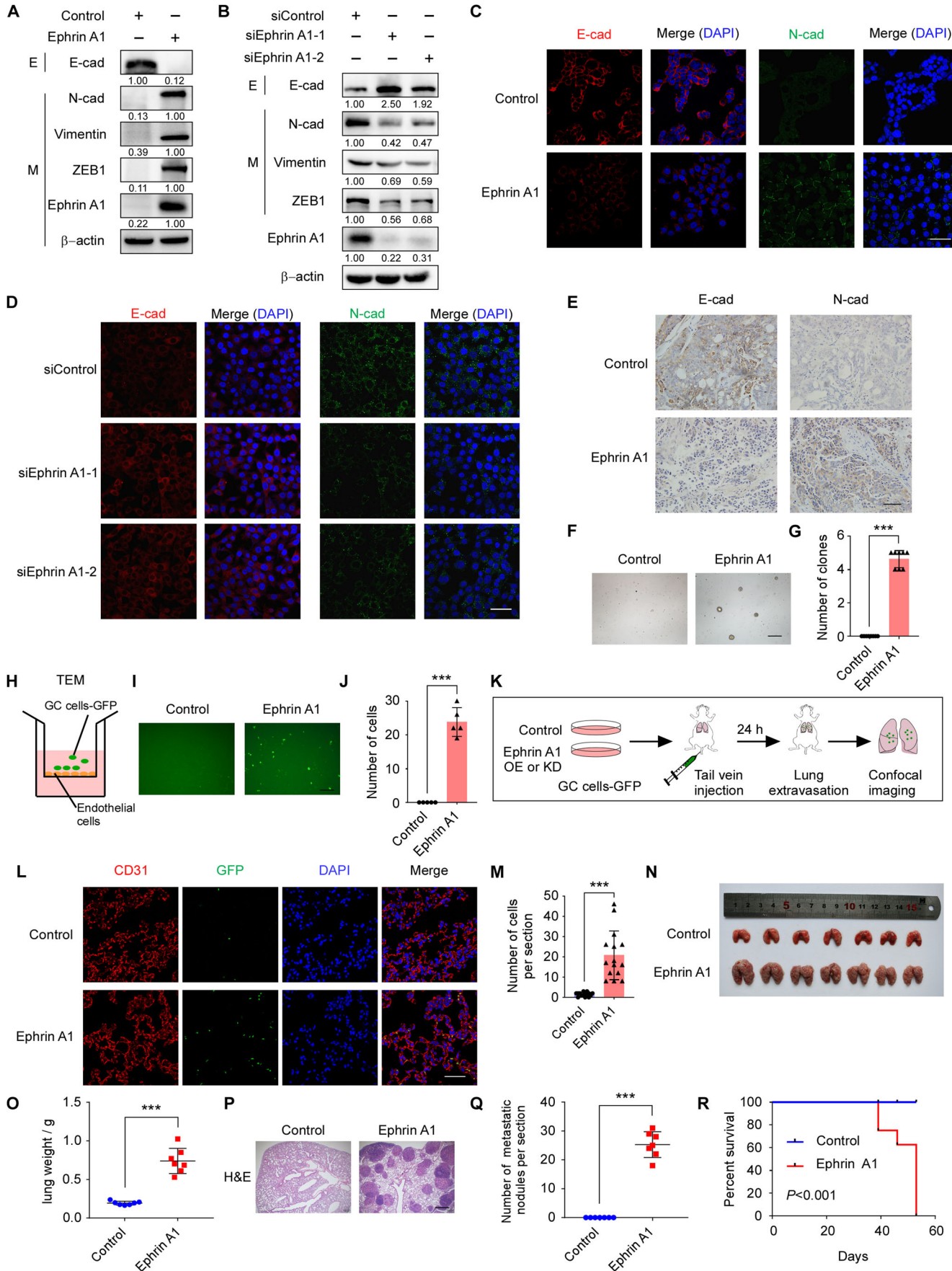

**Figure 1.  Ephrin A1 promotes EMT and metastasis of gastric cancer cells.**

(**A**) Western blot analysis of the expression of EMT markers in NCI-N87 cells transfected with pLVX-Ephrin A1 lentivirus or not. Experiments were performed three times of biological replicates. (**B**) Western blot analysis of EMT markers in MKN45 cells transfected with control or Ephrin A1 siRNAs. Experiments were performed three times of biological replicates. (**C**) Immunofluorescence showed E-cadherin (red) and N-cadherin (green) protein levels in NCI-N87 cells that stably expressed Ephrin A1 or control. DNA was visualized by DAPI. Scale bar, 50 μm. (**D**) Immunofluorescence analysis of control or Ephrin A1-knockdown MKN45 cells stained with E-cadherin (red) and N-cadherin (green) antibodies. DNA was visualized by DAPI. Scale bar, 50 μm. (**E**) Immunohistochemistry analysis of subcutaneous tumors from control or Ephrin A1-overexpressing NCI-N87 cells stained with indicated EMT antibodies. Scale bar, 50 μm. (**F, G**) 3D soft agar colony formation assay of NCI-N87 cells stably expressing Ephrin A1 or not. The number of clones was counted (**G**). $P = 4.41e-13$. Scale bar, 400 μm. Experiments were performed three times of biological replicates. (**H**) Schematic illustration of transendothelial migration (TEM) transwell model in vitro. The upper well was pre-planted with monolayer HUVEC cells and then seeded with GFP- or fluorescent dye-labelled gastric cancer cells to detect the TEM cell number. (**I, J**) TEM analysis of control or Ephrin A1-overexpressiong NCI-N87 cells. The number of TEM cells was counted (**J**). $P = 1.59e-6$. Scale bar, 200 μm. Experiments were performed three times of biological replicates. (**K**) Schematic of the short-term lung colonization assay to analyze the extravasation of gastric cancer cells to lungs in vivo. (**L, M**) Immunofluorescence showed the NCI-N87 cells overexpressing-Ephrin A1 or not extravasated to lungs. The number of extravasated cells was counted (**M**). $P = 1.03e-6$. Scale bar, 50 μm. (**N, O**) Mice were intravenously injected with control or Ephrin A1-overexpressing NCI-N87 cells and subjected to lung metastasis analysis (n = 7 for each group). Lung weights of different groups were measured (**O**). $P = 1.43e-6$. (**P, Q**) H&E staining of cross sections of mice lungs (**P**) and quantification of lung metastases nodules (**Q**). $P = 3.90e-9$. Scale bar, 400 μm. (**R**) Kaplan–Meier survival curve of mice injected of control or Ephrin A1-overexpressing NCI-N87 cells. $P = 0.0008$. E epithelial, M mesenchymal, E-cad E-cadherin, N-cad N-cadherin, OE overexpression, KD knock down, GC gastric cancer. Data are shown as mean ± SD. Statistical significance was determined by Student's t test. \*\*\*$P < 0.001$. Source data are available online for this figure.

different gastric cancer cell lines and found that NCI-N87 and AGS cells had low expression levels of Ephrin A1 and MKN45 cells were highly expressed of Ephrin A1 (Fig. EV1A). Then, we stably overexpressed human Ephrin A1 in NCI-N87 cells by the lentiviral system. Interestingly, NCI-N87 cells overexpressing Ephrin A1 appeared morphological changes, which were loss of tightly clustered features and appearance of elongated mesenchymal features (Fig. EV1B), implying the epithelial-mesenchymal transition (EMT) in gastric cancer cells. Western blot and quantitative RT-PCR analyses further showed that ectopic expression of Ephrin A1 reduced epithelial marker (E-cadherin) expression and increased the expression of mesenchymal markers (N-cadherin, vimentin, transcription factor ZEB1) in NCI-N87 cells (Figs. 1A and EV1C). In addition, downregulation of Ephrin A1 by small interfering RNAs (siRNAs) significantly inhibited EMT process in MKN45 cells (Figs. 1B and EV1D). Similar results were observed in the immunofluorescence staining results (Fig. 1C,D). Moreover, transient overexpression of Ephrin A1 also promoted EMT in AGS cells (Fig. EV1E–G). These data suggest that Ephrin A1 induces EMT of gastric cancer cells.

To further confirm the role of Ephrin A1 in inducing EMT in vivo, we performed subcutaneous injection of control and Ephrin A1-overexpressing NCI-N87 cells to generate tumors. In two weeks, we isolated the tumors and performed immunohistochemical analysis. The results showed that tumors with high Ephrin A1 expression had lower expression of E-cadherin and higher expression of N-cadherin (Fig. 1E).

Since cancer cells undergone EMT can acquire invasive phenotypes to initiate metastatic dissemination (Chaffer and Weinberg, 2011; Mani et al, 2008), so we evaluated whether Ephrin A1 induced the invasive phenotypes of gastric cancer cells. We found that ectopic expression of Ephrin A1 significantly promoted migration and invasion abilities of NCI-N87 (Fig. EV2A–D) and AGS cells (Fig. EV2E–H). In addition, knockdown of Ephrin A1 expression markedly suppressed migration (Fig. EV2I,J) and invasion (Fig. EV2K,L) abilities in MKN45 cells. These results indicate that Ephrin A1 induces EMT to enhance the invasive abilities of gastric cancer cells.

## Ephrin A1 promotes colonization and metastasis of gastric cancer cells

EMT can induce cancer cells enter into a stem-like state which is critical for forming new colonies in metastatic organs (Chaffer and Weinberg, 2011). Then we tested whether Ephrin A1 influenced the colony formation abilities of gastric cancer cells, the results showed that ectopic expression of Ephrin A1 significantly enhanced 3D colony formation of NCI-N87 cells (Fig. 1F,G) and depleted Ephrin A1 expression dramatically reduced the number of colonies in MKN45 cells (Fig. EV3A,B). Transendothelial migration and extravasation are critical steps in the metastatic dissemination of malignant cells (Cheng and Cheng, 2021; Reymond et al, 2013; van Zijl et al, 2011). We investigated whether Ephrin A1 facilitated the extravasation ability of gastric cancer cells and subsequent colonization. An in vitro transendothelial migration transwell assay was performed (Fig. 1H) and the results showed that NCI-N87 cells overexpressing Ephrin A1 exhibited enhanced capacity to transmigrate across HUVECs (human umbilical vein endothelial cells) monolayers (Fig. 1I,J). In contrast, downregulation of Ephrin A1 in MKN45 cells displayed a significantly reduced ability of transendothelial migration (Fig. EV3C,D).

Next, we performed an in vivo lung extravasation assay (Fig. 1K). Green fluorescent protein (GFP) labeled-gastric cancer cells were injected into tail veins of mice. After 24 h, lungs were collected and perfused free of blood with PBS to remove intravascular tumor cells. Immunofluorescence assay showed that ectopic expression of Ephrin A1 significantly improved the extravasation ability of NCI-N87 cells (Fig. 1L,M). Knockdown of Ephrin A1 dramatically decreased the number of MKN45 cells extravasated to lungs (Fig. EV3E,F). Moreover, in vivo lung metastasis assay showed that overexpression of Ephrin A1 in NCI-N87 cells significantly promoted the lung metastasis (Fig. 1N–Q) and reduced overall survival time of mice (Fig. 1R). Knockdown of Ephrin A1 in MKN45 cells by shRNAs (Fig. EV3G) dramatically decreased the lung metastasis abilities (Fig. EV3H–L). Collectively, these data suggest that Ephrin A1 promotes the colonization and metastasis of gastric cancer cells.

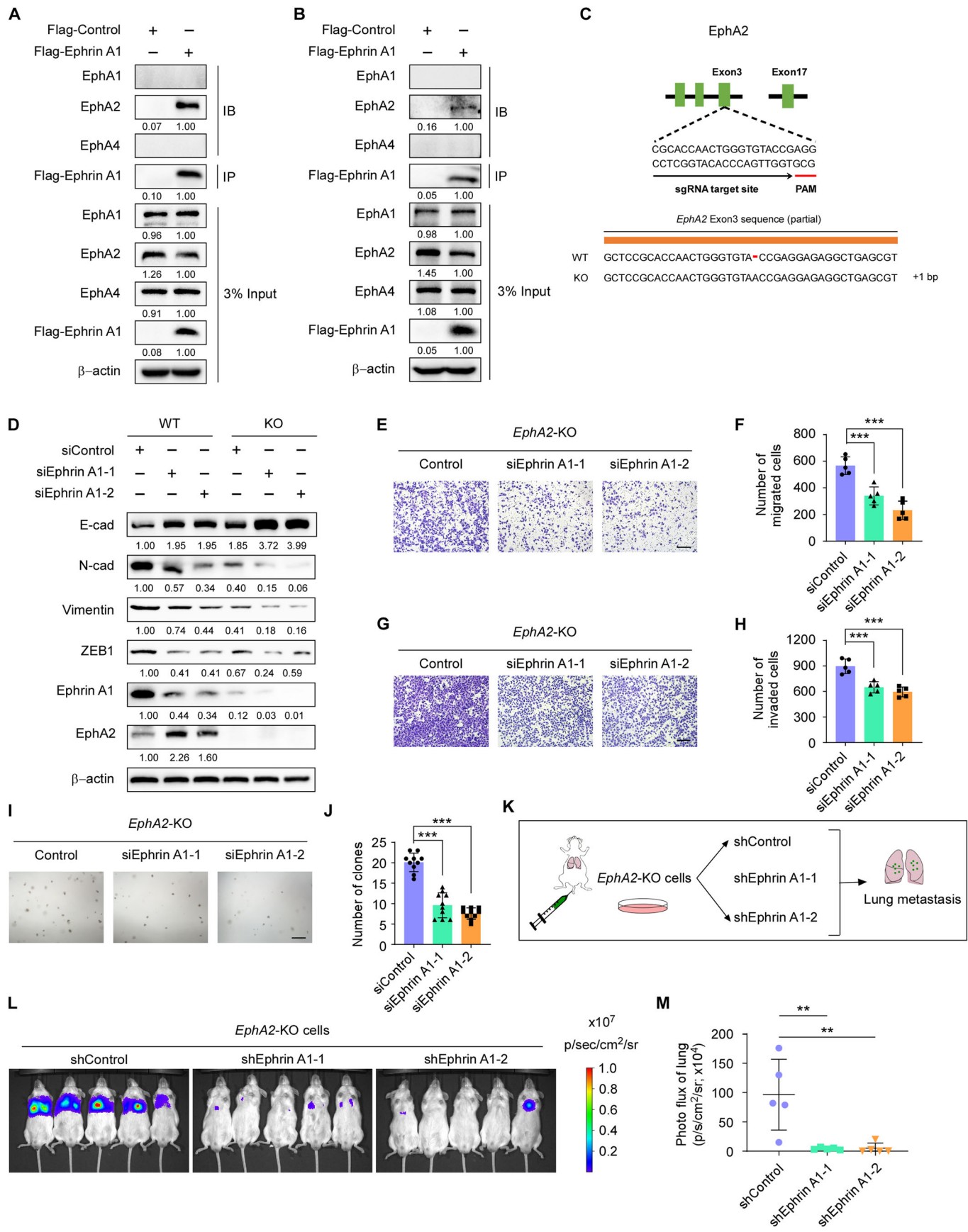

**Figure 2.  Ephrin A1 regulates EMT and metastasis of gastric cancer cells in the EphA2 independent manner.**

(A, B) NCI-N87 (A) or MKN45 (B) cells were transfected with control-Flag or Ephrin A1-Flag plasmids and then applied to immunoprecipitation analysis with anti-Flag beads. The IP samples were subjected to western blot analysis with the indicated antibodies. (C) The construction of *EphA2* knockout (KO) cells by CRISPR-Cas9 system. DNA sequencing was performed to confirm the target site. (D) Western blot analysis of the expression of EMT markers in wild type (WT) or *EphA2*-KO MKN45 cells. (E–H) Transwell migration (E) and invasion (G) analyses of MKN45-*EphA2*-KO cells transfected with control or Ephrin A1 siRNAs. The migrated and invaded cells were counted (F, H). $P$ values from left to right (F), $P = 0.0005$, $P = 1.32e-5$. $P$ values from left to right (H), $P = 0.0004$, $P = 5.74e-5$. Scale bars, 200 μm. Experiments were performed three times of biological replicates. (I, J) 3D soft agar colony formation analysis of control or Ephrin A1-KD MKN45-*EphA2*-KO cells. The number of clones were counted. (J). $P$ values from left to right, $P = 3.88e-10$, $P = 8.14e-12$. Scale bar, 400 μm. Experiments were performed three times of biological replicates. (K) Schematic illustration of the lung metastasis model. Luciferase-labeled control or Ephrin A1-KD MKN45-*EphA2*-KO cells were tail vein injection to NSG mice ($1 \times 10^6$ cells, $n = 5$). (L) In vivo bioluminescence imaging (BLI) of mice were shown. (M) Quantification of BLI signal intensity of mice ($n = 5$ mice per group). $P$ values from left to right, $P = 0.0036$, $P = 0.0039$. Data are shown as mean ± SD. Statistical significance was determined by one-way ANOVA. **$P < 0.01$, ***$P < 0.001$. Source data are available online for this figure.

## Ephrin A1 regulates EMT and metastasis of gastric cancer cells in an EphA2 independent manner

Ephrin A1 is known to be a ligand of Eph receptor tyrosine kinase family (Bartley et al, 1994; Beauchamp and Debinski, 2012). EphA1, EphA2 and EphA4 were reported receptors for Ephrin A1 in different cells (Ende et al, 2014; Ieguchi and Maru, 2019; Wang et al, 2016). To investigate the underlying mechanism of how Ephrin A1 induces EMT and metastasis of gastric cancer cells, we employed immunoprecipitation (IP) assay to identify the interactions of Ephrin A1 with the reported EphA receptors. Our results showed that Ephrin A1 interacted with EphA2 in NCI-N87 (Fig. 2A), MKN45 (Fig. 2B) and AGS (Fig. EV4A) cells, while the interactions between Ephrin A1 and EphA1 or EphA4 were not obvious maybe due to the cell type specificity.

To determine whether EphA2 receptor is required for the functions of Ephrin A1 in regulating EMT and metastasis of gastric cancer cells, we constructed *EphA2* knockout (KO) cell line by CRISPR (clustered regularly interspaced short palindromic repeats)-Cas9 system in MKN45 (Fig. 2C) and AGS cells (Fig. EV4B,C) and tested the EMT and metastasis abilities regulated by Ephrin A1. The results showed that downregulation of Ephrin A1 significantly inhibited EMT process both in wild type (WT) and *EphA2*-KO MKN45 cells (Fig. 2D). Interestingly, *EphA2* KO results in a dramatic reduction in Ephrin A1, which causing significant inhibition of EMT in *EphA2*-KO MKN45 cells compared with WT cells. Furthermore, knockdown of Ephrin A1 expression markedly suppressed migration (Fig. 2E,F), invasion (Fig. 2G,H) and colony formation (Fig. 2I,J) abilities of MKN45-*EphA2*-KO cells, which were consistent with the results in MKN45 wild type cells (Figs. EV2I–L and EV3A,B). In addition, we performed lung metastasis assay to assess the function of Ephrin A1 in the absence of EphA2 in vivo (Fig. 2K). The bioluminescence imaging showed that downregulation of Ephrin A1 in MKN45-*EphA2*-KO cells significantly decreased lung metastasis compared with control cells (Fig. 2L,M).

Moreover, to further investigate whether the functions of Ephrin A1 in inducing EMT and metastasis of gastric cancer cells is EphA2 independent, we ectopically expressed Ephrin A1 in WT and *EphA2*-KO AGS cells and found that Ephrin A1 still significantly promoted EMT process in the absent of EphA2 (Fig. EV4D). In addition, the results of transwell migration and invasion assays showed that ectopic expression of Ephrin A1 also markedly facilitated migration (Fig. EV4E,F) and invasion (Fig. EV4G,H) abilities of AGS-*EphA2*-KO cells, as well as WT AGS cells.

Together, these results demonstrate that the functions of Ephrin A1 in inducing EMT and metastasis of gastric cancer cells are EphA2 receptor independent.

## Ephrin A1 interacts with EGFR

To further explore the underlying mechanism of how Ephrin A1 induces EMT and metastasis of gastric cancer cells, we hypothesize whether Ephrin A1 interacts with other important tyrosine kinase receptors, such as EGFR, FGFR and integrins. Interestingly, we found that Ephrin A1 interacted with EGFR in MKN45 cells (Fig. 3A) and the similar results were detected in NCI-N87 (Fig. 3B) and AGS (Fig. EV4A) cells. In addition, Ephrin A1 interacted with EGFR in the absence of EphA2 (Fig. 3C). Conversely, we also detected that EGFR bound to Ephrin A1 by immunoprecipitation assay (Fig. 3D). Although EGFR has not yet been reported as an Ephrin A1-interacting partner, it has been known to be important for gastric cancer progression (Arienti et al, 2019; Zhang et al, 2017).

To identify the regions of EGFR required for Ephrin A1 binding, we generated a series of deletion constructs of EGFR (EGFR-△ECD, extracellular domain deletion; EGFR-△D1, domain I deletion; EGFR-△D3, domain III depletion; EGFR-△D1 + 3, domain I and III depletion), and examined the binding ability of EGFR to Ephrin A1 (Fig. 3E). Compared with wild type EGFR, the interactions between EGFR-△D1 and EGFR-△D3 with Ephrin A1 were significantly reduced; EGFR-△ECD and EGFR-△D1 + 3 exhibited almost loss of binding to Ephrin A1 (Fig. 3F) in MKN45 cells. Together, these results suggest that domains I and III of EGFR ECD, which were known to bind to EGF (Li et al, 2005), were also required for Ephrin A1 binding.

## Ephrin A1 activates EGFR as a new functional ligand

The above results prompted us to ask whether Ephrin A1 acts as an EGFR ligand. To experimentally test this hypothesis, we firstly performed pull-down assay in vitro and confirmed that Ephrin A1 and EGFR bound to each other directly (Fig. 4A,B). Next, we evaluated the activation of EGFR by Ephrin A1. Western blot assays showed that ectopic expression of Ephrin A1 significantly increased the phosphorylation (Fig. 4C) and dimerization (Fig. 4D) levels of EGFR, and induced EGFR downstream molecules (Keller and Schmidt, 2017) activation (Fig. 4E) in NCI-N87 cells. Consistently, ectopic expression of Ephrin A1 significantly facilitated the phosphorylation (Fig. EV5A), dimerization (Fig. EV5D) and downstream signaling activation (Fig. EV5G) of EGFR in AGS

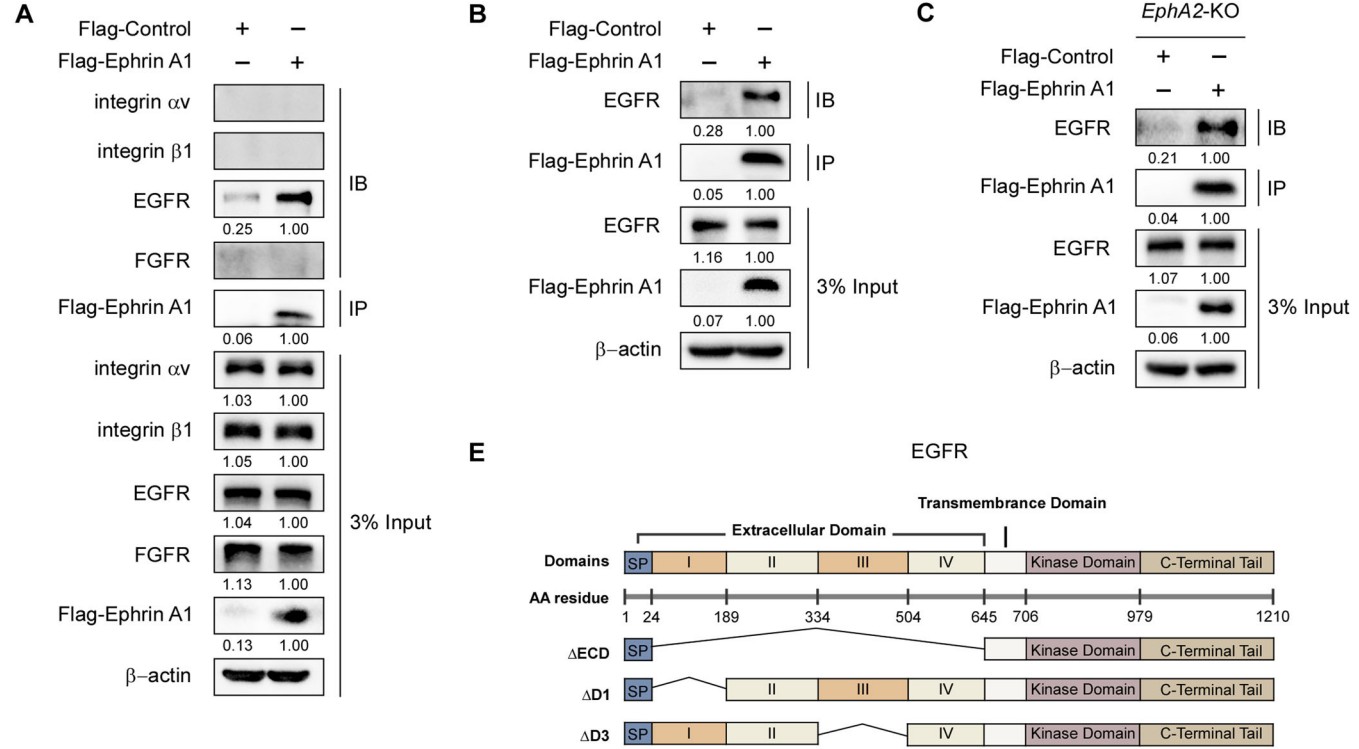

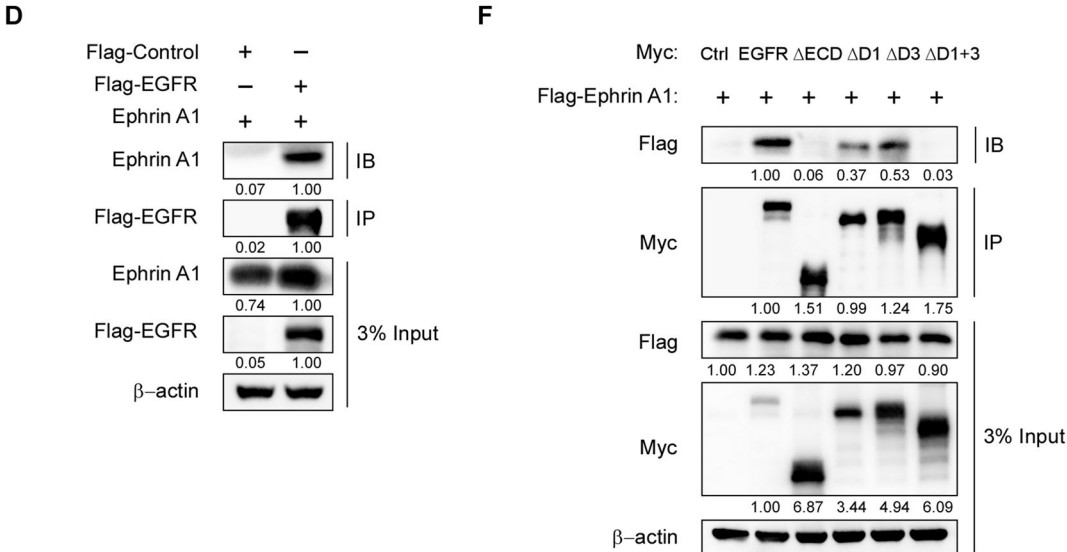

**Figure 3. Ephrin A1 interacts with EGFR.**

(A–C) MKN45 (A), NCI-N87 (B) and MKN45-*EphA2*-KO (C) cells were transfected with control-Flag or Ephrin A1-Flag plasmids and then applied to immunoprecipitation analysis with anti-Flag beads. The IP samples were subjected to western blot analysis with the indicated antibodies. (D) Ephrin A1-overexpressing MKN45 cells were transfected with control-Flag or EGFR-Flag vectors and subjected to immunoprecipitation analysis. The IP samples were subjected to western blot analysis with the indicated antibodies. (E) Schematic illustration of EGFR mapping domains. (F) MKN45 cells transfected with different EGFR mapping plasmids were subjected to immunoprecipitation analysis with anti-Myc beads. The IP samples were subjected to western blot analysis with the indicated antibodies. Experiments were performed three times of biological replicates. Source data are available online for this figure.

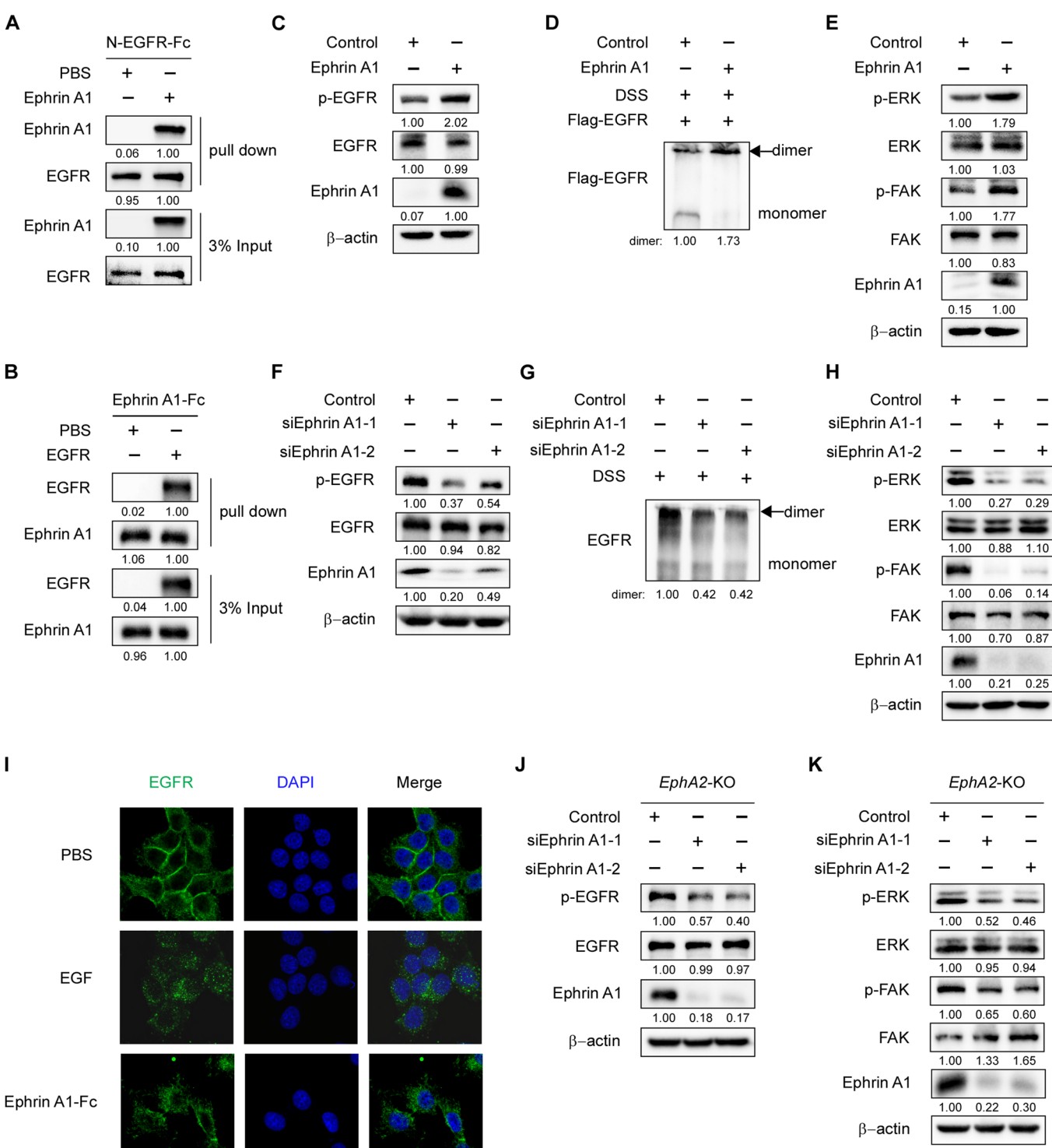

cells. In addition, Ephrin A1-Fc protein treatment also increased EGFR phosphorylation (Fig. EV5B,C) and dimerization (Fig. EV5E,F) levels in AGS cells. In contrast, downregulation of Ephrin A1 in MKN45 cells significantly inhibited EGFR phosphorylation (Fig. 4F), dimerization (Fig. 4G) and downstream signaling activation levels (Fig. 4H). Similar to EGF, the most known ligand of EGFR, we observed the increasing internalization of EGFR into

the cytoplasm after Ephrin A1-Fc protein treatment in MKN45 cells using confocal fluorescence microscopy (Fig. 4I). Collectively, these results demonstrate that Ephrin A1 serves as a new EGFR ligand to activate EGFR signaling.

To further validate that Ephrin A1 activates EGFR signaling independent of EphA2, we detected the phosphorylation levels of EGFR and downstream signaling in *EphA2* KO cells. The results

**Figure 4. Ephrin A1 acts as a ligand of EGFR.**

(A) N-EGFR-Fc and Ephrin A1 proteins were incubated in lysis buffer overnight at 4 °C. Protein A/G beads were then used for pull-down assay. (B) Pull-down assay was performed with Ephrin A1-Fc and EGFR proteins. (C) Western blot analysis of phosphorylation level of EGFR in control or Ephrin A1-overexpressing NCI-N87 cells. (D) Control or Ephrin A1-overexpressing NCI-N87 cells were collected in PBS. DSS was added to a final concentration of 2.5 mM and incubated on ice for 2 h. The reaction was quenched by 10 mM Tris-HCl for 15 min on ice. Finally, the cells were lysed and analyzed by western blot. (E) Western blot analysis of the phosphorylation levels of ERK and FAK with indicated antibodies. (F) MKN45 cells were transfected with control or Ephrin A1 siRNAs and then applied to western blot analysis with the indicated antibodies. (G) MKN45 cells were transfected with control or Ephrin A1 siRNAs and then collected the cells in PBS. DSS was added to a final concentration of 2.5 mM and incubated on ice for 2 h. The reaction was quenched by 10 mM Tris-HCl for 15 min on ice. Finally, the cells were lysed and analyzed by western blot. (H) Western blot analysis of the phosphorylation levels of ERK and FAK in MKN45 cells transfected with control or Ephrin A1 siRNAs. (I) Immunofluorescence analysis of MKN45 cells pre-treated with PBS, 100 ng/ml EGF or 5 µg/ml Ephrin A1 proteins with EGFR antibody. DNA was visualized by DAPI. Scale bar, 20 µm. (J, K) Western blot analysis of phosphorylation levels of EGFR, FAK and ERK in MKN45-EphA2-KO cells transfected with control or Ephrin A1 siRNAs. Experiments were performed three times of biological replicates. Source data are available online for this figure.

showed that ectopic expression of Ephrin A1 still significantly activated EGFR signaling in AGS-*EphA2*-KO cells (Fig. EV5H,I), while downregulation of Ephrin A1 in MKN45-*EphA2*-KO cells also significantly suppressed the phosphorylation levels of EGFR and downstream signaling (Fig. 4J,K). These data indicate that Ephrin A1 activating EGFR signaling is independent of EphA2.

## Ephrin A1 induces EMT of gastric cancer cells through EGFR signaling

To determine whether EGFR signaling mediates the function of Ephrin A1 in regulating EMT of gastric cancer cells, we generated an *EGFR* knockout cell line by CRISPR-Cas9 system in MKN45 cells (Fig. 5A). Western blot indicated that knockdown of Ephrin A1 failed to induce E-cadherin expression in the *EGFR*-KO cells (Fig. 5B). In addition, we treated cells with U0126 (ERK inhibitor) or defactinib (FAK inhibitor) and detected the EMT process induced by Ephrin A1. The results showed that U0126 or defactinib treatment significantly suppressed the EMT process (Fig. 5C) and inhibited the 3D colony formation (Fig. 5D,E) and transendothelial migration (Fig. 5F,G) abilities induced by Ephrin A1 in NCI-N87 cells. Moreover, blockage of ERK and FAK activation also inhibited EMT (Fig. EV6A), migration (Fig. EV6B,C) and invasion (Fig. EV6D,E) abilities induced by Ephrin A1 in AGS cells. Together, these results suggest that Ephrin A1 may induce EMT of gastric cancer cells through activating EGFR signaling.

## Blocking EGFR activation by Erlotinib inhibits the function of Ephrin A1 in gastric cancer cells

To further interrupt the cancer-promoting functions of Ephrin A1, we inhibited EGFR activation by EGFR tyrosine kinase inhibitor (TKI) erlotinib and then evaluated the function of Ephrin A1. We found that erlotinib treatment significantly suppressed the activation of EGFR signaling in NCI-N87 cells (Fig. 5H). Furthermore, we evaluated the 3D colony formation and transendothelial migration abilities of NCI-N87 cells and found that erlotinib treatment significantly inhibited the functions of Ephrin A1 (Fig. 5I–L). In addition, the EMT (Fig. EV6F), migration (Fig. EV6G,H) and invasion (Fig. EV6I,J) abilities induced by Ephrin A1 were significantly suppressed in AGS cells treatment with erlotinib.

Next, we performed an in vivo lung metastasis assay. Mice injected with Ephrin A1-overexpressing NCI-N87 cells were subjected to erlotinib treatment or control vehicle three times per week (Fig. 6A). Strikingly, mice with Ephrin A1-overexpressing cancer cells developed severe metastatic lesions, while erlotinib treatment strongly reduced the lung metastases of gastric cancer cells (Fig. 6B–E). These findings suggest that Ephrin A1 promotes metastasis of gastric cancer cells through EGFR activation.

## Ephrin A1 expression level is significantly correlated with EGFR activation in gastric cancer patients

To further address the clinical relevance between Ephrin A1 expression level and EGFR phosphorylation level in gastric cancer patients, we performed a human gastric tumor tissue array analysis. Immunohistochemistry staining showed that Ephrin A1 protein levels were positively correlated with the activation status of EGFR in gastric cancer patients (Fig. 7A,B).

Moreover, gastric cancer patients with high Ephrin A1 expression levels or high EGFR phosphorylation levels had poor prognosis compared to low expression group (Fig. 7C,E). Further multivariate Cox analysis showed that high expression level of Ephrin A1 was an independent positive prognostic factor for predicting outcomes of gastric cancer patients (HR: 2.26; $P < 0.05$) (Fig. 7D). These results indicate that Ephrin A1 may be a biomarker for predicting prognosis of gastric cancer patients.

# Discussion

Gastric cancer patients with metastases have very poor outcomes. However, the mechanisms of gastric cancer metastasis still remain elusive. Here, our data showed that Ephrin A1 bound to EGFR and activated EGFR signaling as a new EGFR ligand. Ephrin A1 further induced EMT and facilitated invasion, transendothelial migration, colonization and metastasis capacities of gastric cancer cells. Consistent with our findings, Ephrin A1 expression level was positively correlated with EGFR phosphorylation level in gastric cancer patients. Importantly, EGFR inhibitor erlotinib treatment suppressed metastatic ability of Ephrin A1 in mice, indicating potential therapeutic application of Ephrin A1/EGFR signaling (Fig. 7F).

EGFR is one of the four members of ErbB family of tyrosine kinase receptors. Seven well-known ligands are responsible for the activation of EGFR including EGF, transforming growth factor-α (TGF-α), amphiregulin, heparin-binding EGF-like growth factor, betacellulin, epiregulin, epigen and neuregulin (Chen et al, 2016; Roskoski, 2014; Yewale et al, 2013), which have various affinities to

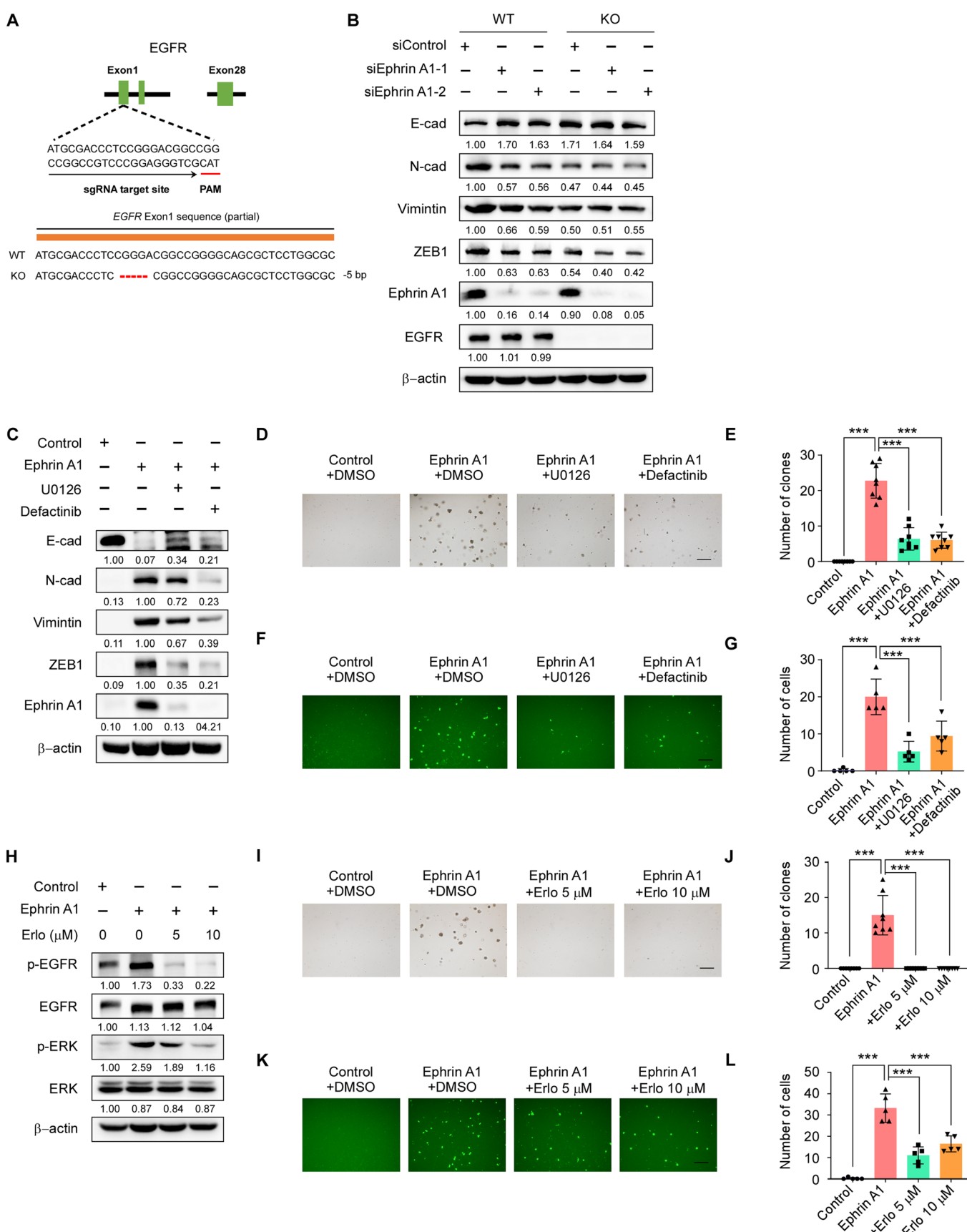

**Figure 5.  Ephrin A1 promotes EMT and metastatic abilities of gastric cancer cells through EGFR signaling in vitro.**

(A) The construction of *EGFR* knockout cells by the CRISPR-Cas9 system. DNA sequencing was performed to confirm the target site. (B) Western blot analysis of the expression of EMT markers in WT or *EGFR* knockout (KO) MKN45 cells. (C) Ephrin A1-overexpressing NCI-N87 cells were treated with 20 μM U0126 or 5 μM defactinib separately for 24 h, and applied to western blot analysis with EMT antibodies. (D, E) The 3D soft agar colony formation ability was analyzed in control or Ephrin A1-overexpressing NCI-N87 cells that treated with 10 μM U0126 and 1 μM defactinib or not. The number of clones was counted (E). *P* values from left to right, $P = 7e{-}14$, $P = 1.99e{-}10$, $P = 1.19e{-}10$. Scale bar, 400 μm. (F, G) TEM analysis in control or Ephrin A1-overexpressing NCI-N87 cells that treated with 10 μM U0126 and 1 μM defactinib or not. The number of TEM cells was counted (G). *P* values from left to right, $P = 5.44e{-}7$, $P = 2.26e{-}5$, $P = 0.0009$. Scale bar, 200 μm. (H) Western blot analysis of EGFR and ERK phosphorylation levels in control, Ephrin A1-overexpressing, and erlotinib-treated Ephrin A1-overexpressing NCI-N87 cells with indicated antibodies. (I, J) The 3D soft agar colony formation ability was analyzed in control or Ephrin A1-overexpressing NCI-N87 cells treated with DMSO or erlotinib. The number of clones was counted (J). *P* values from left to right, $P = 1.01e{-}10$, $P = 1.01e{-}10$, $P = 1.01e{-}10$. Scale bar, 400 μm. (K, L) TEM analysis of control or Ephrin A1-overexpressing NCI-N87 cells treated with DMSO or erlotinib. The number of TEM cells was counted (L). *P* values from left to right, $P = 1.11e{-}8$, $P = 2.61e{-}6$, $P = 7.83e{-}5$. Scale bar, 200 μm. Erlo, Erlotinib. Experiments were performed three times of biological replicates. Data are shown as mean ± SD. Statistical significance was determined by one-way ANOVA. ****P* < 0.001. Source data are available online for this figure.

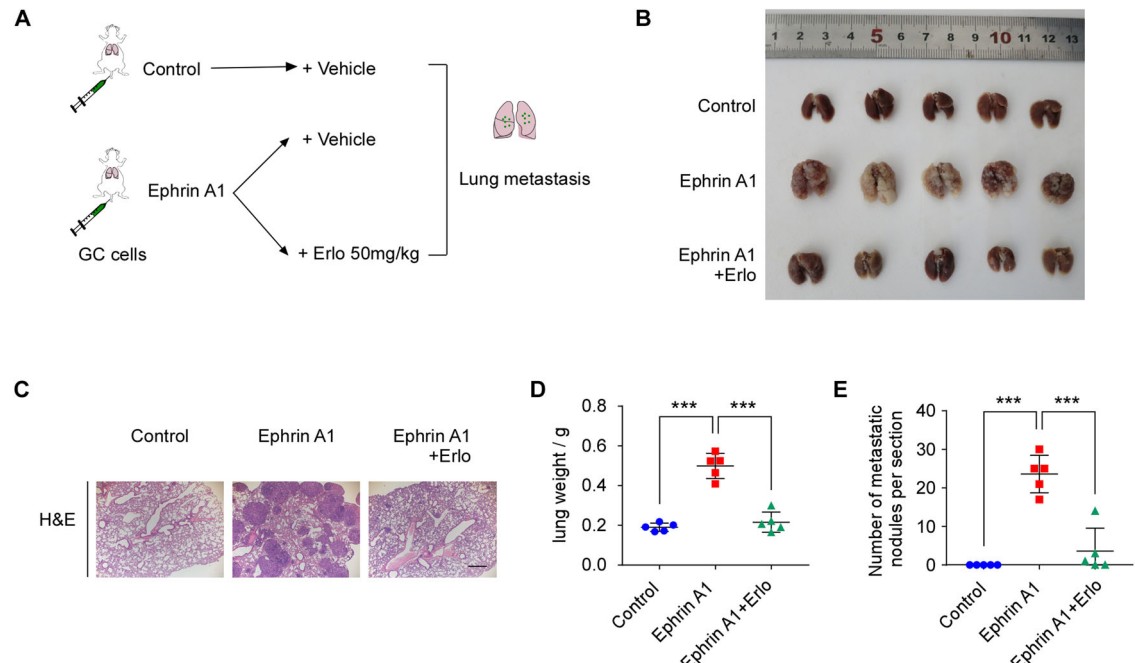

**Figure 6.  Ephrin A1 promotes metastasis of gastric cancer cells through EGFR activation in vivo.**

(A) Mice were intravenously injected with control or Ephrin A1-overexpressing NCI-N87 cells and then treated with erlotinib or vesicle ($n = 5$ for each group). (B, C) Lungs were collected and subjected to lung metastasis analysis. H&E staining of cross sections of mice lungs was shown (C). Scale bars, 400 μm. (D) Lung weights of each group were measured ($n = 5$ mice per group). *P* values from left to right, $P = 9.18e{-}7$, $P = 2.30e{-}6$. (E) Metastatic nodules per section of mice lungs were counted ($n = 5$ mice per group). *P* values from left to right, $P = 6.26e{-}6$, $P = 3.33e{-}5$. Erlo, Erlotinib. Data are shown as mean ± SD. Statistical significance was determined by one-way ANOVA. ****P* < 0.001. Source data are available online for this figure.

EGFR (Sanders et al, 2013). And recent studies reported angiogenin is a new EGFR ligand and contribute to pancreatic cancer progression (Wang et al, 2018). EGFR was aberrantly activated in many cancers and was an important drug target for cancer therapies; however, almost all patients who initially benefited from EGFR-targeted therapies eventually developed resistance (Chong and Jänne, 2013; Guardiola et al, 2019; Sigismund et al, 2018). In addition, EGFR targeting therapies produced little effect in gastric cancer patients (Lordick et al, 2013; Waddell et al, 2013). Further understanding of the molecular mechanisms of EGFR signaling is crucial in the treatment of cancers. Here, we found that Ephrin A1 was a ligand of EGFR and activated EGFR signaling, which may offer new possibilities for EGFR activation. The co-crystal structure

between Ephrin A1 and EGFR interaction is important and worthwhile to be further explored. It may help to explain why the domains characterized in this study are required and potentially facilitate the development of small molecules specifically targeting this binding.

Ephrin A1 has been reported to play crucial roles in many cancers, including gastric cancer (Beauchamp and Debinski, 2012; Zhuo et al, 2019). And it has also been reported to be a secreted protein (Alford et al, 2010; Ieguchi et al, 2014; Wykosky et al, 2008). However, the mechanisms in regulating cancer metastasis remains largely unknown. Although EphA2, the primary receptor of Ephrin A1, has been reported to induce EMT of cancer cells (Fattet et al, 2020; Huang et al, 2014), the function of Ephrin A1 in

**A**

High Ephrin A1 High p-EGFR Y1068 Low Ephrin A1 Low p-EGFR Y1068

case1 case2 200um

**B**

r=0.5998
$P<0.0001$
n=98

(scatter plot) H-score of p-EGFR vs H-score of Ephrin A1

**C**

Ephrin A1 low (n=15) high (n=79)

$P$ value = 0.0357

5-year survival probability (%) vs Time (months)

**D**

| Parameters | | HR (95%CI) | $P$-value |
|---|---|---|---|
| Age (>=60 vs <60) | | 1.48 (0.96-2.27) | 7.65e-02 |
| Gender (Male vs Female) | | 0.94 (0.64-1.40) | 7.66e-01 |
| AJCC M (M1 vs M0) | | 2.59 (1.36-4.91) | 3.64e-03 |
| AJCC Stage (III-IV vs I-II) | | 0.79 (0.25-2.49) | 6.90e-01 |
| Ephrin A1 (High vs Low) | | 2.26 (1.11-4.60) | 2.48e-02 |

0.25 0.50 1.0 2.0 4.0

Ephrin A1

**E**

p-EGFR low (n=60) high (n=34)

$P$ value = 0.0327

5-year survival probability (%) vs Time (months)

**F**

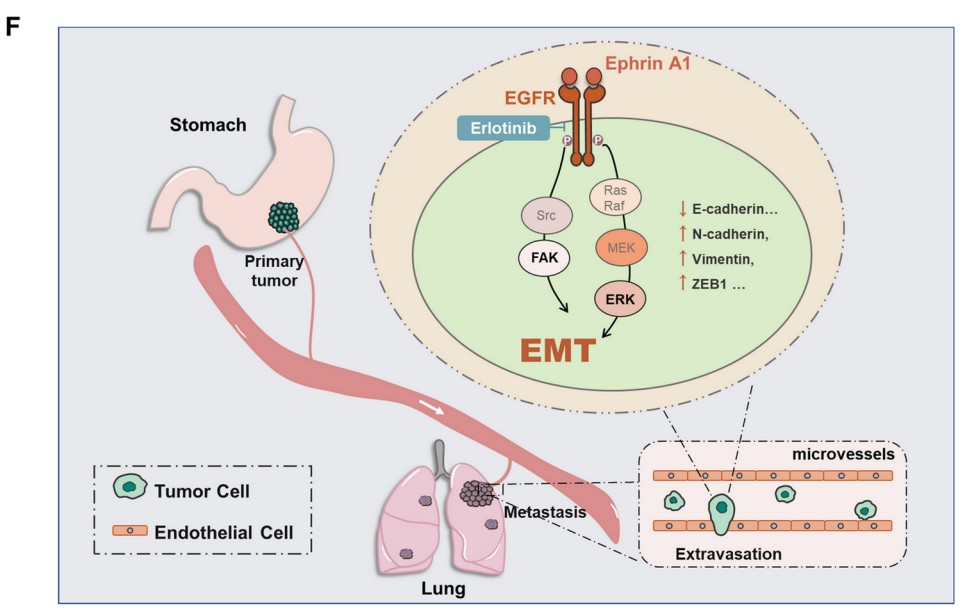

**Figure 7. Pathological relevance of Ephrin A1 expression and EGFR activation in gastric cancer patients.**

(A) Immunohistochemical analysis of human gastric cancer tissue array probed with anti-Ephrin A1 and anti-p-EGFR (Y1068) antibodies. Representative images of different immunohistochemical staining were shown. (B) The expression correlation between Ephrin A1 and p-EGFR was analyzed by Pearson correlation analysis. $P = 6.72e\text{-}11$. (C) Kaplan–Meier analysis of the correlation between Ephrin A1 levels with overall survival in gastric cancer tissues. $P = 0.0357$. (D) Multivariate Cox analysis of individual prognostic factors for gastric cancer patients in the tissue array. All the bars correspond to 95% CI. $P$ values were calculated by Multivariate Cox Proportional-Hazards Model. (E) Kaplan–Meier analysis of the correlation between p-EGFR levels with overall survival in gastric cancer tissues. $P = 0.0327$. (F) Working model of Ephrin A1 activating EGFR signaling to facilitate EMT and metastasis of gastric cancer cells. AJCC American Joint Committee on Cancer, CI confidence interval, HR hazard ratio. Data are shown as mean ± SD. Correlations between two variables were analyzed by Pearson correlation (B). Comparisons between two Kaplan–Meier curves were analyzed by the long-rank test (C, E). $P < 0.05$ was considered statistically significant. Source data are available online for this figure.

cancer EMT was unconfirmed. In this study, we found the function of Ephrin A1 in inducing EMT was independent of EphA2 although *EphA2* KO dramatically decreases Ephrin A1 levels. Interestingly, we uncovered that Ephrin A1 induced EMT of gastric cancer cells by activating EGFR signaling to promote metastatic capacities, which provides a new mechanistic insight for gastric cancer metastasis.

Although we demonstrated Ephrin A1 functioned as the ligand of EGFR to induce EMT and metastasis in gastric cancer cells in this study, Ephrin A1 was reported to be highly expressed in multiple tumors and EGFR was an important target in many cancers. Therefore, it is worthwhile to further investigate the function of Ephrin A1-EGFR axis in other tumor types.

# Methods

## Reagents and tools table

| Reagent/resource | Reference or source | Identifier or catalog number |
| --- | --- | --- |
| **Experimental models** | | |
| NCI-N87 cells | ATCC | CRL-5822 |
| AGS cells | ATCC | CRL-1739 |
| MKN45 cells | The Chinese Academy of Medical Sciences (Beijing, China) | 1101HUM-PUMC000229 |
| Tissue array | Shanghai Outdo Biotech (Shanghai, China) | HStmA180Su15 |
| SCID mice | Shanghai Model Organisms Center | SM-015 |
| NSG mice | Shanghai Model Organisms Center | NM-NSG-001 |
| **Recombinant DNA** | | |
| pcDNA3.1 | Invitrogen | V79020 |
| pcDNA3.1/myc-His | Invitrogen | V80020 |
| pCS2-Flag | Addgene | 16331 |
| pLVX-Puro | Clontech | 632164 |
| pLKO.1 | Addgene | 8453 |
| pcDNA3.1-Ephrin A1 | This paper | N/A |
| pCS2-Ephrin A1-Flag | This paper | N/A |
| pLVX-Ephrin A1 | This paper | N/A |
| pcDNA5-EGFR-Flag | This paper | N/A |
| pcDNA3.1-EGFR-Myc | This paper | N/A |
| pcDNA3.1-EGFR-ΔECD- Myc | This paper | N/A |
| pcDNA3.1-EGFR-ΔD1-Myc | This paper | N/A |
| pcDNA3.1-EGFR-ΔD3-Myc | This paper | N/A |
| pcDNA3.1-EGFR-ΔD1 + 3-Myc | This paper | N/A |
| pLKO.1-shEphrin A1-1 | This paper | N/A |

| Reagent/resource | Reference or source | Identifier or catalog number |
| --- | --- | --- |
| pLKO.1-shEphrin A1-2 | This paper | N/A |
| **Antibodies** | | |
| Ephrin A1 Rabbit pAb | ABclonal | Cat# A5341 |
| E-cadherin Mouse mAb | Proteintech | Cat# 60335-1-Ig |
| E-Cadherin Rabbit mAb | CST | Cat# 3195 |
| N-Cadherin Rabbit mAb | CST | Cat# 13116 |
| Vimentin Rabbit pAb | Proteintech | Cat# 10366-1-AP |
| Vimentin Rabbit mAb | CST | Cat# 5741 |
| ZEB1 Rabbit pAb | Proteintech | Cat# 21544-1-AP |
| ZEB1 Rabbit mAb | CST | Cat# 70512 |
| EphA2 Rabbit mAb | CST | Cat# 6997 |
| EGF Receptor Rabbit mAb | CST | Cat# 4267 |
| EGFR Rabbit mAb | ABclonal | Cat# A23381 |
| Phospho-EGF Receptor (Tyr1068) Rabbit mAb | CST | Cat# 3777 |
| Flag-tag Rabbit pAb | Proteintech | Cat# 20543-1-AP |
| FAK Mouse mAb | Diagbio | Cat# db6186 |
| FAK Rabbit mAb | CST | Cat# 71433 |
| Phospho-FAK (Tyr397) Rabbit pAb | Diagbio | Cat# db2584 |
| Phospho-FAK (Tyr397) Rabbit mAb | CST | Cat# 8556 |
| ERK1/2 Rabbit pAb | Diagbio | Cat# db758 |
| ERK1/2 Rabbit mAb | CST | Cat# 4695 |
| Phospho-ERK1/2 Rabbit pAb | Diagbio | Cat# db269 |
| Phospho-Erk1/2 Rabbit mAb | CST | Cat# 4377 |
| EphA1 Rabbit pAb | Proteintech | Cat# 18698-1-AP |
| EphA4 Rabbit pAb | Proteintech | Cat# 21875-1-AP |
| FGFR1 Rabbit pAb | Diagbio | Cat# db3221 |
| Myc-Tag Mouse mAb | CST | Cat# 2276 |
| β-actin Rabbit mAb | ABclonal | Cat# AC026 |
| Alexa Fluor 488 conjugated secondary antibody | Invitrogen | Cat# A-11008 |
| Alexa Fluor 555 conjugated secondary antibody | Invitrogen | Cat# A-21431 |
| **Oligonucleotides and other sequence-based reagents** | | |
| siControl (5'-3') | UUCUCCGAAC GUGUCACGU | N/A |
| siEphrin A1-1 (5'-3') | GAAGGACACA GCUACUACUAC | N/A |
| siEphrin A1-2 (5'-3') | AGCAUGAAGA CCGCUGCUU | N/A |
| EphA2 gRNA | CGCACCAACTG GGTGTACCGAGG | N/A |
| EGFR gRNA | ATGCGACCCTC CGGGACGGCCGG | N/A |

| Reagent/resource | Reference or source | Identifier or catalog number |
|---|---|---|
| *Ephrin A1*-qPCR-primer F | CTACTACATCT CCAAACCCATCC | N/A |
| *Ephrin A1*-qPCR-primer R | CTGGGTCATCT GCTGCAAGTC | N/A |
| *E-cadherin*-qPCR-primer F | AGGCCAAGCAG CAGTACATT | N/A |
| *E-cadherin*-qPCR-primer R | ATTCACATCCA GCACATCCA | N/A |
| *N-cadherin*-qPCR-primer F | AGGTTTGCCAG TGTGACTCC | N/A |
| *N-cadherin*-qPCR-primer R | TGATGATGCAG AGCAGGATG | N/A |
| *Vimentin*-qPCR-primer F | AGCTAACCAAC GACAAAGCC | N/A |
| *Vimentin*-qPCR-primer R | TCCACTTTGCGT TCAAGGTC | N/A |
| *ZEB1*-qPCR-primer F | CCTGTCCATATT GTGATAGAGGC | N/A |
| *ZEB1*-qPCR-primer R | ACCCAGACTGCG TCACATGT | N/A |
| *GADPH*-qPCR-primer F | GGAGCGAGATCC CTCCAAAAT | N/A |
| *GADPH*-qPCR-primer R | GGCTGTTGTCATA CTTCTCATGG | N/A |
| pLKO- Ephrin A1-sh1-F | CCGGCGTCTTCTG GAACAGTTCAAAC TCGAGTTTGAACTG TTCCAGAAGACGT TTTT | N/A |
| pLKO- Ephrin A1-R | AATTAAAAACGTC TTCTGGAACAGTT CAAACTCGAGTTTG AACTGTTCCAGAA GACG | N/A |
| pLKO- Ephrin A1-F | CCGGAGAGGTGCG GGTTCTACATAGCT CGAGCTATGTAGAA CCCGCACCTCTTTTTTG | N/A |
| pLKO- Ephrin A1-R | AATTCAAAAAAG AGGTGCGGGTTCT ACATAGCTCGAGC TATGTAGAAC CCGCACCTCT | N/A |
| **Chemicals, enzymes and other reagents** | | |
| Erlotinib | MCE | HY-50896 |
| U0126 | MCE | HY-12031A |
| Defactinib | MCE | HY-12289 |
| Ephrin A1-Fc protein | R&D Systems | 6417-A1 |
| EGF protein | MCE | HY-P7109 |
| N-EGFR-Fc protein | R&D Systems | 344-ER |
| EGFR protein | R&D Systems | 1095-ER |
| DSS | ThermoFisher | A39267 |
| **Software** | | |
| GraphPad Prism 7 | https://www.graphpad.com/ | N/A |
| ImageJ | https://imagej.net/ij/ | N/A |
| **Other** | | |
| Transwell | Corning | 3422 |

## Cell lines and cell culture

The human gastric cancer cell lines NCI-N87 and AGS were purchased from American Type Culture Collection (ATCC; Manassas, VA, USA). MKN45 cells were obtained from the Chinese Academy of Medical Sciences (Beijing, China). NCI-N87 and MKN45 cells were maintained in RPMI-1640 medium (Corning, USA) containing 10% fetal bovine serum (FBS; ExCell, China) and 1% penicillin/streptomycin (Gibco, 15140) at 37 °C with 5% $CO_2$. AGS cells were cultured in F-12 medium (Corning, USA) containing 10% FBS and 1% penicillin/streptomycin at 37 °C with 5% $CO_2$. All cell lines were tested to be negative for mycoplasma.

The inhibitors and proteins used in cell treatment including Erlotinib (HY-50896, MCE), U0126 (HY-12031A, MCE), Defactinib (HY-12289, MCE), Ephrin A1-Fc protein (6417-A1, R&D Systems).

## Human specimens

Clinical samples for tissue array analysis in this study were obtained from Shanghai Outdo Biotech (Shanghai, China). The study was permitted by the ethics committee of Shanghai Outdo Biotech (Approval Number: SHYJS-CP-1801009).

## Mouse models

The animal experiments were approved by the Institutional Animal Care and Use Committee (IACUC) of Zhejiang University. SCID mice and NSG mice (female, 4–5 weeks old) and were bred in pathogen-free environment in the Animal Facility of Zhejiang University. For lung metastasis assay, NCI-N87 cells ($2 \times 10^6$) were injected into tail veins of SCID mice. After 6 weeks, lungs were isolated from mice and the number of lung metastatic nodules per section was counted after hematoxylin and eosin (H&E) staining. For tumor growth assay, $5 \times 10^6$ NCI-N87 cells mixed with Matrigel (BD Biosciences, USA) at a ratio of 1:1 were injected subcutaneously into mice and the tumors were isolated and subjected to immunohistochemical staining. For short-term lung colonization assay, NCI-N87-GFP cells were injected into tail veins of mice. After 24 h, lungs were perfused free of blood with PBS and sectioned for immunofluorescence assay. For erlotinib treatment assay, SCID mice were intravenously injected with $2 \times 10^6$ NCI-N87 cells and erlotinib (50 mg/kg solved in vehicle) or control vehicle was administrated one day after cells injection by tail veins three times per week. Lungs were dissected and the metastatic lesions were counted. For lung metastasis assays of Ephrin A1 down-regulation, MKN45 cells ($5 \times 10^5$) or MKN45-*EphA2*-KO cells ($1 \times 10^6$) were injected into tail veins of NSG mice. The lung metastasis was detected by bioluminescence imaging and the number of lung metastatic nodules was counted after hematoxylin and eosin (H&E) staining. Mice were randomly grouped before different treatments.

## RNA extraction and quantitative RT-PCR

Total RNA was extracted using TRIzol reagent (Invitrogen, USA) and reversely transcribed into complementary DNA using HiScript II Reverse Transcriptase (Vazyme, China). Quantitative RT-PCR was then performed using ChamQ SYBR qPCR Master Mix (Vazyme, China) on a LightCycler 480 instrument (Roche, Switzerland). The data were analyzed as described previously (Livak and Schmittgen, 2001). All reactions were performed at least three times.

## Plasmids construction, transfection and lentivirus package

For transient overexpression, full-length human Ephrin A1 cDNA was amplified by PCR and cloned into pcDNA3.1 and PCS2-Flag vectors as previously described (Zhuo et al, 2019). PcDNA5-EGFR-Flag plasmid was a kind gift from Dr. Jimin Shao's lab (Zhejiang University). Full-length human EGFR cDNA was cloned into pcDNA 3.1/Myc-His C vector. Transient plasmid transfection was carried out with PolyJet reagent (SignaGen, USA) according to the manufacturer's protocol.

For stable overexpression, full-length human Ephrin A1 cDNA was cloned into pLVX-Puro vector. PLVX-Ephrin A1 plasmid and packaging plasmids were then transfected into HEK-293T cells using PolyJet reagent. After 48 h, the viral medium was collected, filtered and concentrated to infect the target cells. The efficiency of overexpression was assessed by western blotting.

For stable knockdown, shRNAs targeting Ephrin A1 were designed and cloned into the pLKO.1 vector. HEK293T cells were transfected with the indicated lentivirus expression vector and packaging plasmids. After 48 h, the viral medium was collected, filtered and concentrated to infect the target cells. After 24 h infection, cells were treated with 1 mg/mL of puromycin (Thermo Scientific) for 1 week. The efficiency of downregulation was assessed by western blotting.

The siRNAs were synthesized (GenePharma, China) and transfected into cells using GenMute reagent (SignaGen, USA) according to the manufacturer's protocol. The efficiency of transient knockdown was assessed by western blotting.

## Migration and invasion assays

For cell migration assay, $5 \times 10^4$ cells were suspended in culture medium containing 1% FBS and seeded in the upper chamber of Transwell (8-μm pore, Corning, USA). The lower chambers were filled with 700 μl culture medium containing 10% FBS. After indicated times, the migrated cells were stained with 0.1% crystal violet (Sigma, USA) and quantified. For cell invasion assay, the upper chamber was pre-coated with a 50 μl mixture of Matrigel (BD Biosciences, USA) and RPMI-1640 with a ratio of 1:9 for 2 h at 37 °C. The following steps were the same with migration assay.

## Transendothelial migration assay

In all, $1 \times 10^5$ HUVECs were plated in the upper chamber of the Transwell insert and grown to be confluent. Then $5 \times 10^4$ gastric cancer-GFP cells were resuspended in 1% FBS culture medium and plated on the top of the endothelial monolayer. The lower chambers were filled with culture medium containing 10% FBS. Cells were allowed to migrate for indicated times at 37 °C in 5% $CO_2$. Cells on the inner side of each insert were scraped. Cells that migrated to the basolateral side of the membrane were visualized with immunofluorescent microscope. Pictures of five random fields were captured for quantification.

## 3D soft agar colony formation assay

Agarose and indicated medium were preheated to 37 °C in water bath. 1.3% agarose (A600015, Sangon Biotech) and $2 \times 1640$ (CR-31802, Cienry) were mixed in a ratio of 1:1 and supplemented with 10% FBS and 1% penicillin/streptomycin solution. In all, 600 μl mixture was added into the bottom layer of 24-well plate per well and then put the plate in 4 °C to solidify. Cells were resuspended in $2 \times 1640$ medium and counted. Then 0.6% agarose and $2 \times 1640$ containing cells were mixed in a ratio of 1:1 and supplemented with 10% FBS and 1% penicillin/streptomycin solution. In total, 600 μl cell/agar mixture was added to the upper layer of the 24-well plates with a final concentration of 2000 cells/well. After growing in the 37 °C incubator for 2 weeks, the number of colonies was counted under microscopy.

## Dimerization assay

Gastric cancer cells were transfected with control or pcDNA3.1-Ephrn A1 plasmid and starved in serum-free medium for 24 h, or cells were starved for 24 h and then treated with PBS, 100 ng/ml EGF or 1 μg/ml Ephrin A1-Fc for 30 min. The cells were then collected in 0.5 ml PBS and crosslinked on ice for 2 h by disuccinimidyl suberate (DSS: 2.5 mM, ThermoFisher, USA). 1 M Tris-HCl buffer (pH 7.5) was then added to stop the cross-linking reactions at a final concentration of 10 mM and incubated for 15 min on ice. Cells were then lysed for 20 min on ice and EGFR dimerization was analyzed by western blotting.

## Western blot assay and immunoprecipitation

For western blot assay, cells were lysed in cold RIPA buffer (P0013B, Beyotime) containing protein inhibitor cocktail (Roche). Protein samples were separated by SDS-PAGE and incubated with indicated antibodies. The signals were detected by the ChemiDoc Touch Imaging System (Bio-Rad, USA).

For immunoprecipitation (IP) experiment, cells were lysed in lysis buffer (20 mM Tris pH 8.0, 150 mM NaCl, 1 mM EDTA, 1 mM EGTA, 1% Triton-100) with protease inhibitor cocktail and then incubated on ice for 30 min. Cell lysates were incubated with anti-Flag or anti-Myc antibodies and protein A/G-agarose beads (Santa Cruz). The immunoprecipitates were subsequently analyzed by western blot.

## Pull-down assay

Pull-down assays were performed as described previously (Lu et al, 2017). N-EGFR-Fc (344-ER, R&D Systems) protein and Ephrin A1 protein purified from bacteria were incubated in lysis buffer overnight at 4 °C. Protein A/G magnetic beads (HY-K0202, MCE) were then used for pull-down assay. The reverse pull-down assay was done with EGFR protein (1095-ER, R&D Systems) and Ephrin A1-Fc protein (6417-A1, R&D Systems).

## Immunofluorescence and immunohistochemistry

Immunofluorescence experiments were performed as described previously (Zhou et al, 2003). Gastric cancer cells seeded on coverslips were fixed with 4% formaldehyde for 15 min. After blocking in 3% BSA for 30 min, the coverslips were incubated with primary antibodies against E-cadherin (60335-1-Ig, Proteintech) and N-cadherin (13116, CST). Then cells were washed and incubated with Alexa Fluor 488 conjugated secondary antibody (A-11008, Invitrogen) and Alexa Fluor 555 conjugated secondary antibody (A-21431, Invitrogen) for 1 h. DNA was stained with DAPI (Sigma). The coverslips were mounted in ProLong Gold

Antifade Reagent (Life Technologies). Images were captured by confocal fluorescence microscopy with an oil immersion ×60 objective (OLYMPUS IX83-FV3000-OSR, Olympus, Japan).

For immunofluorescence of mouse lung tissues, tissues were dehydrated by gradient sucrose solutions and embedded into OCT compound (Sakura, Tokyo, Japan). The OCT-embedded tissues were then cut into frozen sections and used for immunofluorescence staining.

For immunohistochemical staining, the formalin-fixed subcutaneous tumors or mouse lung tissues were embedded with paraffin. The paraffin-embedded tumors were then sectioned, stained with antibodies of E-cadherin (60335-1-Ig, Proteintech) and N-cadherin (13116, CST), and imaged with a microscope (Olympus). The paraffin-embedded mouse lung tissues were then sectioned, stained with H&E, and imaged with a microscope.

## The CRISPR/Cas9 system

*EphA2* and *EGFR* knockout cell lines were performed by the CRISPR-Cas9 genomic editing system, as previously reported (Cong et al, 2013). The EphA2 and EGFR gRNAs were then cloned into the vector Pep-330x and transfected into MKN45 cells. The AGS *EphA2* KO cells were generated by lentiCRISPRv2 plasmid system. The monoclonal cells were selected for DNA sequencing and western blot to identify knockout cell lines.

## Tissue array analysis

For tissue array assay, the gastric cancer tissue array was obtained from the Shanghai Outdo Biotech and contains 98 paired of gastric cancer samples. The gastric cancer tissue array was stained with anti-Ephrin A1 antibody (A5341, ABclonal) and anti-p-EGFR (Y1068) antibody (3777, CST). The staining results were analyzed with a Panoramic MIDI tissue array scanner (3D HISTECH). Ephrin A1 and p-EGFR (Y1068) expression levels were determined by histochemistry score (H-score) of the staining signal.

## Statistical analysis

Unpaired two-tailed Student's $t$ test was used to compare differences between two experimental groups. One-way analysis of variance (ANOVA) with Tukey's test was used to compare differences more than two groups. Results are presented as mean ± standard deviation. Correlations between two variables were performed by Pearson correlation. Comparisons between two Kaplan–Meier curves were analyzed by the long-rank test. Forest plots showing the multivariable analysis of prognostic parameters for overall survival were determined using multivariate Cox regression model. The high and low groups were determined by the optimal cutoff algorithm which was defined as described previously (Budczies et al, 2012). All protein band shows in western blots were quantified by ImageJ software and normalized to β-actin. GraphPad Prism 7 software was used to perform statistical analysis. $P < 0.05$ was considered statistically significant.

## Data availability

No data amenable to large-scale repository deposition were generated in this study.

The source data of this paper are collected in the following database record: biostudies:S-SCDT-10_1038-S44318-025-00363-x.

## Peer review information

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

## Acknowledgements

We thank Yuehong Yang, Xiaoyi Yan, Xiangrui Liu, Shanshan Xie, Lu Zhang, Yuliang Huang and other lab members for technical assistance and helpful discussions. Thanks Junli Xuan for the technical supports from the Core Facilities, Zhejiang University School of Medicine. This work was supported by the National Natural Science Foundation of China (U21A20197, 82173040, 81972276), the Fundamental Research Funds for the Central Universities (2022QZJH47, K20240166), and the Postdoctoral Fellowship Program of CPSF under Grant Number GZC20232281.

## Author contributions

**Shuang Li**: Data curation; Formal analysis; Validation; Investigation; Visualization; Methodology; Writing—original draft; Writing—review and editing. **Meng Sun**: Data curation; Formal analysis; Validation; Writing—review and editing. **Yun Cui**: Data curation; Validation; Investigation. **Dongyang Guo**:

Formal analysis; Validation; Investigation; Methodology. **Feng Yang**: Data curation; Formal analysis; Methodology. **Qiang Sun**: Formal analysis; Validation; Investigation. **Yinuo Ding**: Formal analysis; Investigation. **Mengjie Li**: Formal analysis; Methodology. **Yiman Liu**: Investigation; Methodology. **Guangshuo Ou**: Writing—review and editing. **Wei Zhuo**: Conceptualization; Data curation; Supervision; Funding acquisition; Writing—review and editing. **Tianhua Zhou**: Conceptualization; Resources; Supervision; Funding acquisition; Writing—review and editing.

Source data underlying figure panels in this paper may have individual authorship assigned. Where available, figure panel/source data authorship is listed in the following database record: biostudies:S-SCDT-10_1038-S44318-025-00363-x.

## Disclosure and competing interests statement

The authors declare no competing interests.

# Expanded View Figures

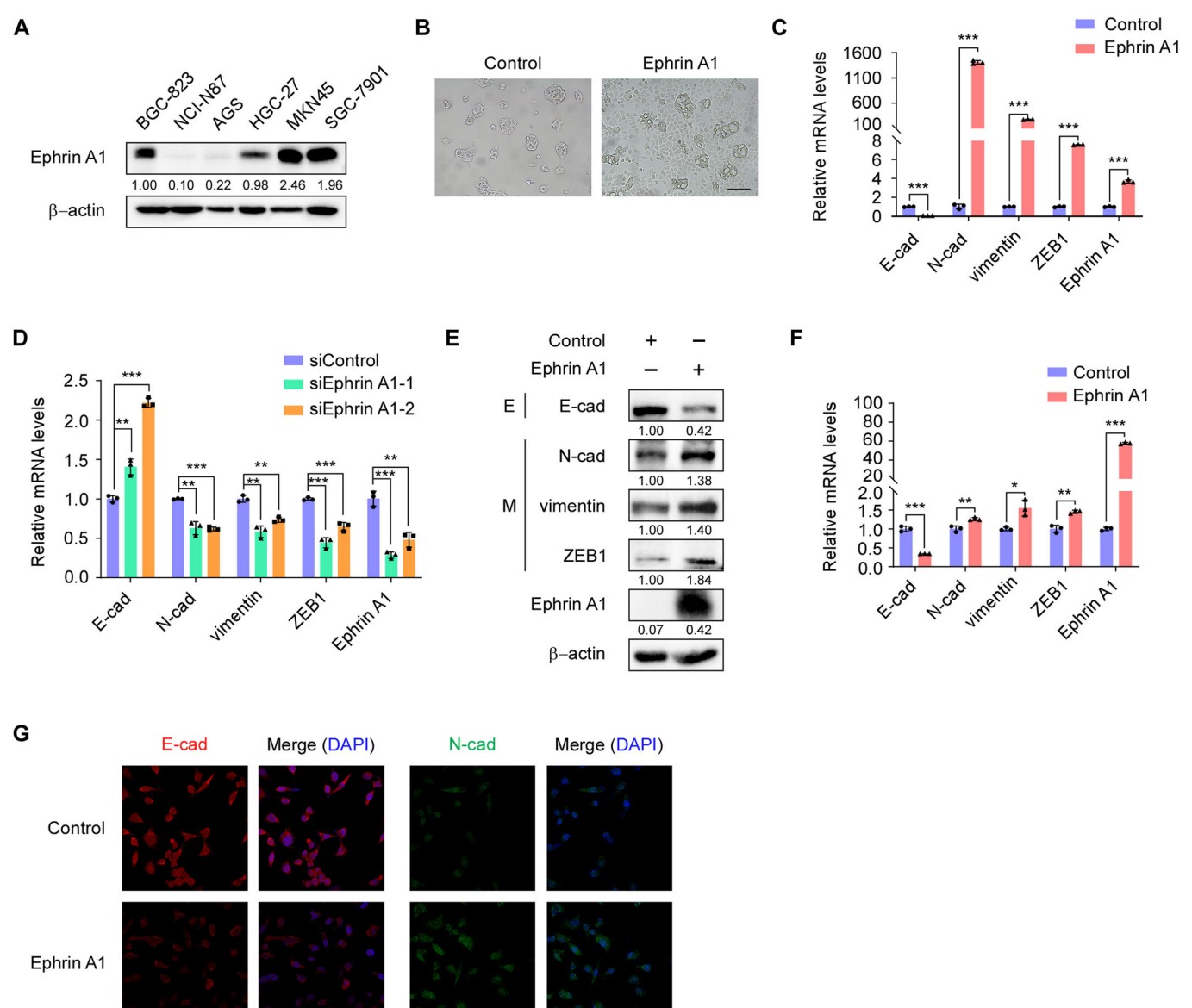

**Figure EV1.  Ephrin A1 promotes EMT of gastric cancer cells.**

(**A**) Western blot analysis of the expression levels of Ephrin A1 in different gastric cancer cells. (**B**) Brightfield images showing the morphological changes in NCI-N87 cells stably expressing Ephrin A1. Scale bar, 200 μm. (**C**) Quantitative RT-PCR analysis of the expression levels of EMT markers in NCI-N87 cells transfected with pLVX-Ephrin A1 lentivirus or not. *P* values from left to right, *P* = 1.33e-6, *P* = 2.65e-7, *P* = 1.63e-7, *P* = 7.89e-9, *P* = 1.38e-5. (**D**) Quantitative RT-PCR analysis of EMT markers in MKN45 cells transfected with control or Ephrin A1 siRNAs. *P* values from left to right, *P* = 0.0029, *P* = 8.13e-6, *P* = 0.0015, *P* = 1.33e-5, *P* = 0.0011, *P* = 0.0010, *P* = 0.0001, *P* = 0.0004, *P* = 0.0003, *P* = 0.0026. (**E**, **F**) Western blot and quantitative RT-PCR analyses of the expression of EMT markers in control and Ephrin A1-overexpressing AGS cells. *P* values from left to right, *P* = 7.92e-5, *P* = 0.0068, *P* = 0.0111, *P* = 0.0017, *P* = 6.69e-8. (**G**) Immunofluorescence staining of EMT markers in control and Ephrin A1-overexpressing AGS cells. DNA was visualized by DAPI. Scale bar, 50 μm. Experiments were performed three times of biological replicates. Data are shown as mean ± SD. *\*P* < 0.05, *\*\*P* < 0.01, *\*\*\*P* < 0.001 (Student's *t* test). Source data are available online for this figure.

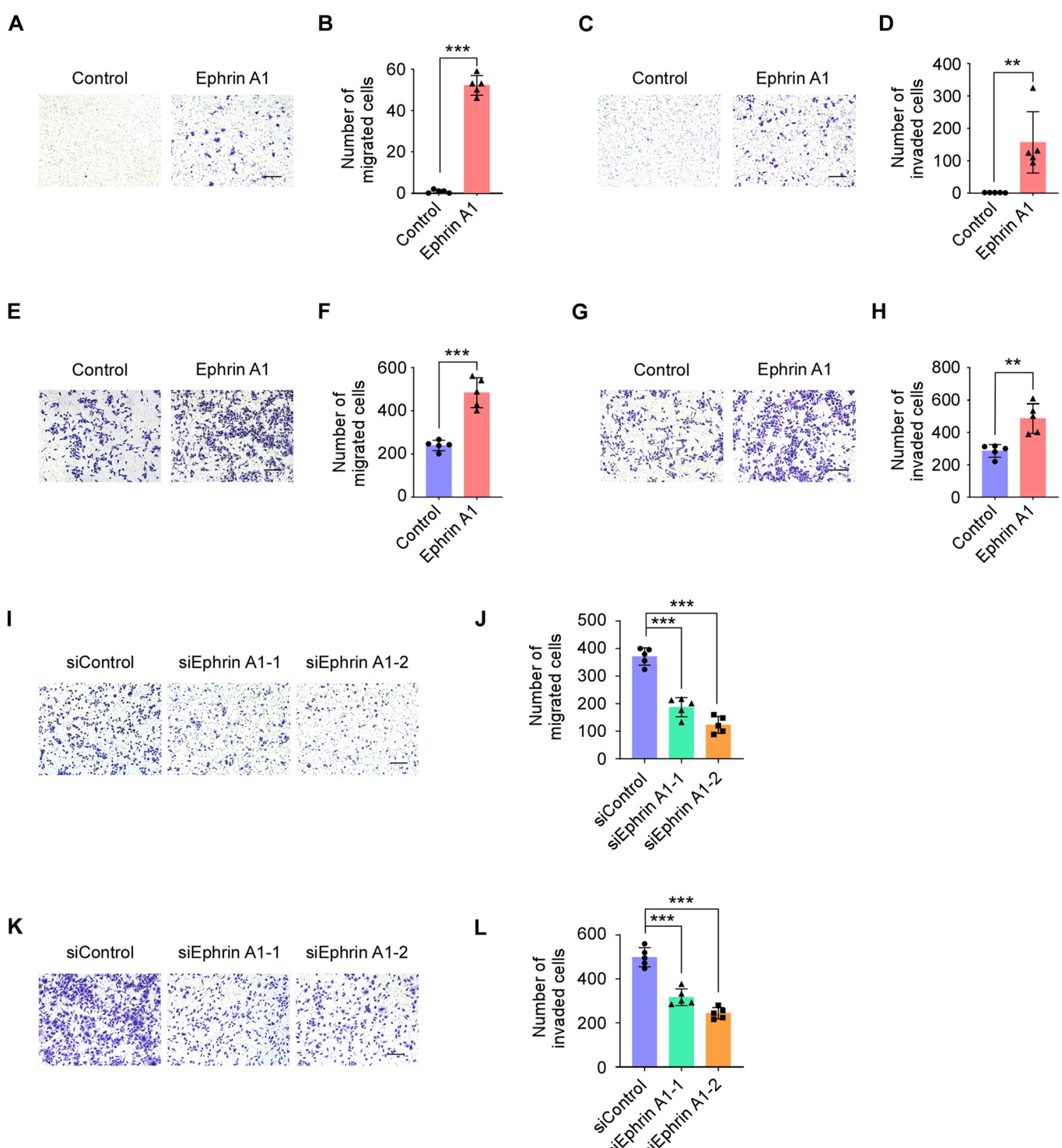

**Figure EV2. Ephrin A1 promotes migration and invasion abilities of gastric cancer cells.**

(A–D) Transwell migration and invasion assays were performed in NCI-N87 cells stably expressing Ephrin A1 or not. The migrated and invaded cells were counted (B, $P = 1.05e$-8; D, $P = 0.0064$). Scale bars, 200 μm. (E–H) Transwell migration and invasion analyses of control and Ephrin A1-overexpressing AGS cells. The migrated and invaded cells were counted (F, $P = 7.27e$-5; H, $P = 0.0021$). Scale bars, 200 μm. (I–L) Transwell migration and invasion analyses of MKN45 cells transfected with control or Ephrin A1 siRNAs. The migrated and invaded cells were counted (J, L). $P$ values from left to right (J), $P = 3.05e$-6, $P = 1.19e$-7. $P$ values from left to right (L), $P = 1.13e$-5, $P = 3.19e$-7. Scale bars, 200 μm. Experiments were performed three times of biological replicates. Data are shown as mean ± SD. Statistical significance was determined by Student's $t$ test (B, D, F, H) and one-way ANOVA (J, L). **$P < 0.01$, ***$P < 0.001$. Source data are available online for this figure.

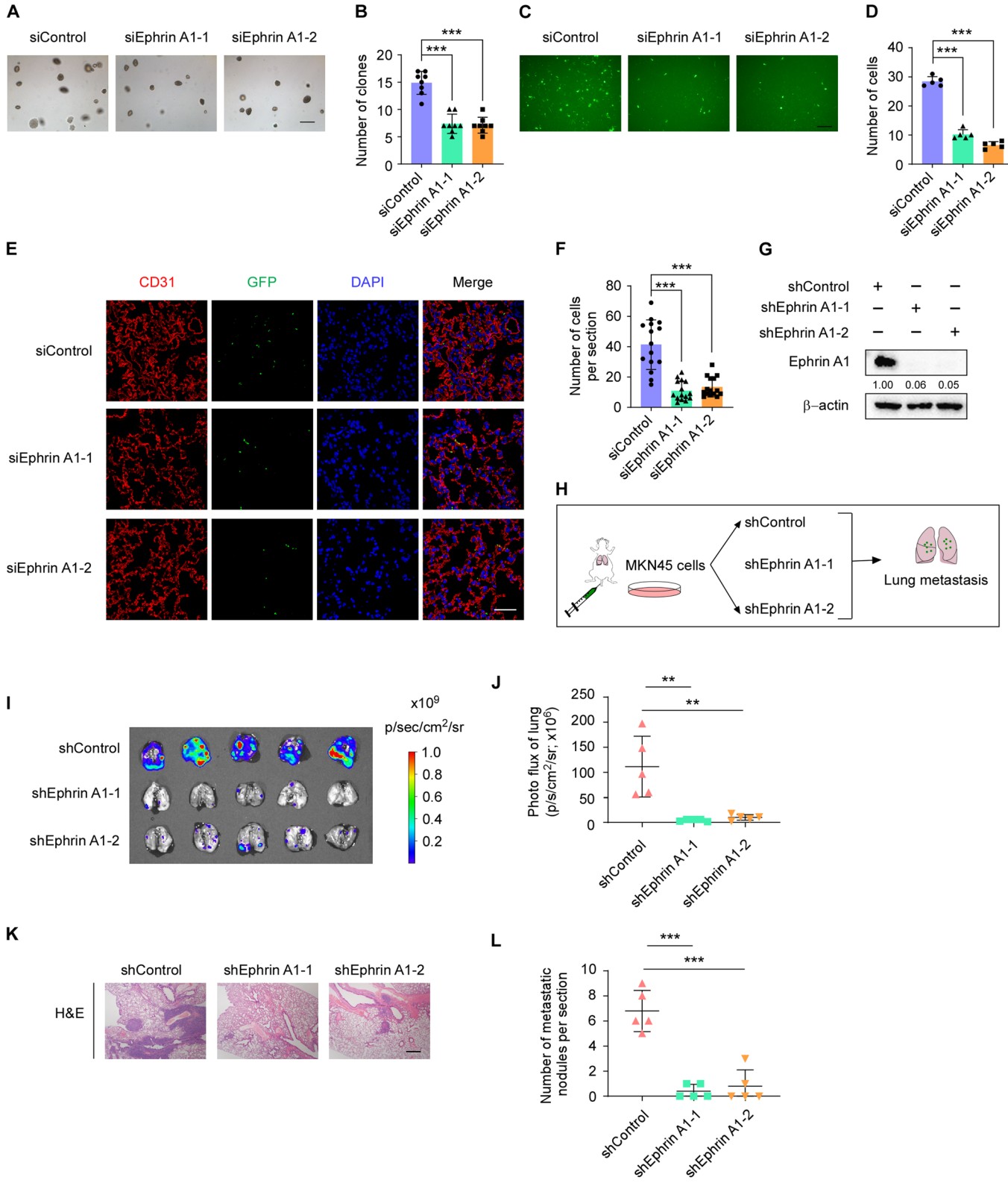

◀ **Figure EV3. Downregulation of Ephrin A1 inhibits colonization and metastasis abilities of gastric cancer cells.**

(A, B) 3D soft agar colony formation analysis of control or Ephrin A1 knockdown MKN45 cells. The number of clones was counted (B). $P$ values from left to right, $P = 1.18\text{e-}7$, $P = 6.90\text{e-}8$. Scale bar, 400 µm. Experiments were performed three times of biological replicates. (C, D) TEM analysis of MKN45 cells transfected with control or Ephrin A1 siRNAs. The number of migrated cells was counted (D). $P$ values from left to right, $P = 7.44\text{e-}10$, $P = 7.29\text{e-}11$. Scale bar, 200 µm. Experiments were performed three times of biological replicates. (E, F) Immunofluorescence analysis of the control or Ephrin A1 knockdown MKN45 cells that extravasated to lungs. The number of extravasated cells was counted (F). $P$ values from left to right, $P = 4.17\text{e-}9$, $P = 3.39\text{e-}8$. Scale bar, 50 µm. (G) The expression of Ephrin A1 was detected by western blot assay in MKN45 and MKN45-Ephrin A1 KD cells. (H) Schematic representation of the in vivo lung metastasis model. Luciferase-labeled MKN45 and MKN45-Ephrin A1 KD cells were tail vein injection to NSG mice ($5 \times 10^5$ cells, $n = 5$). (I) In vivo bioluminescence imaging (BLI) of lungs are shown. (J) Quantification of BLI signal intensity of lungs ($n = 5$ mice per group). $P$ values from left to right, $P = 0.0012$, $P = 0.0018$. (K, L) H&E staining of cross sections of mice lungs (K) and quantification of lung metastases nodules (L). $N = 5$ in each group. $P$ values from left to right, $P = 9.37\text{e-}6$, $P = 1.80\text{e-}5$. Scale bar, 400 µm. Data are shown as mean ± SD. Statistical significance was determined one-way ANOVA. $**P < 0.01$, $***P < 0.001$. Source data are available online for this figure.

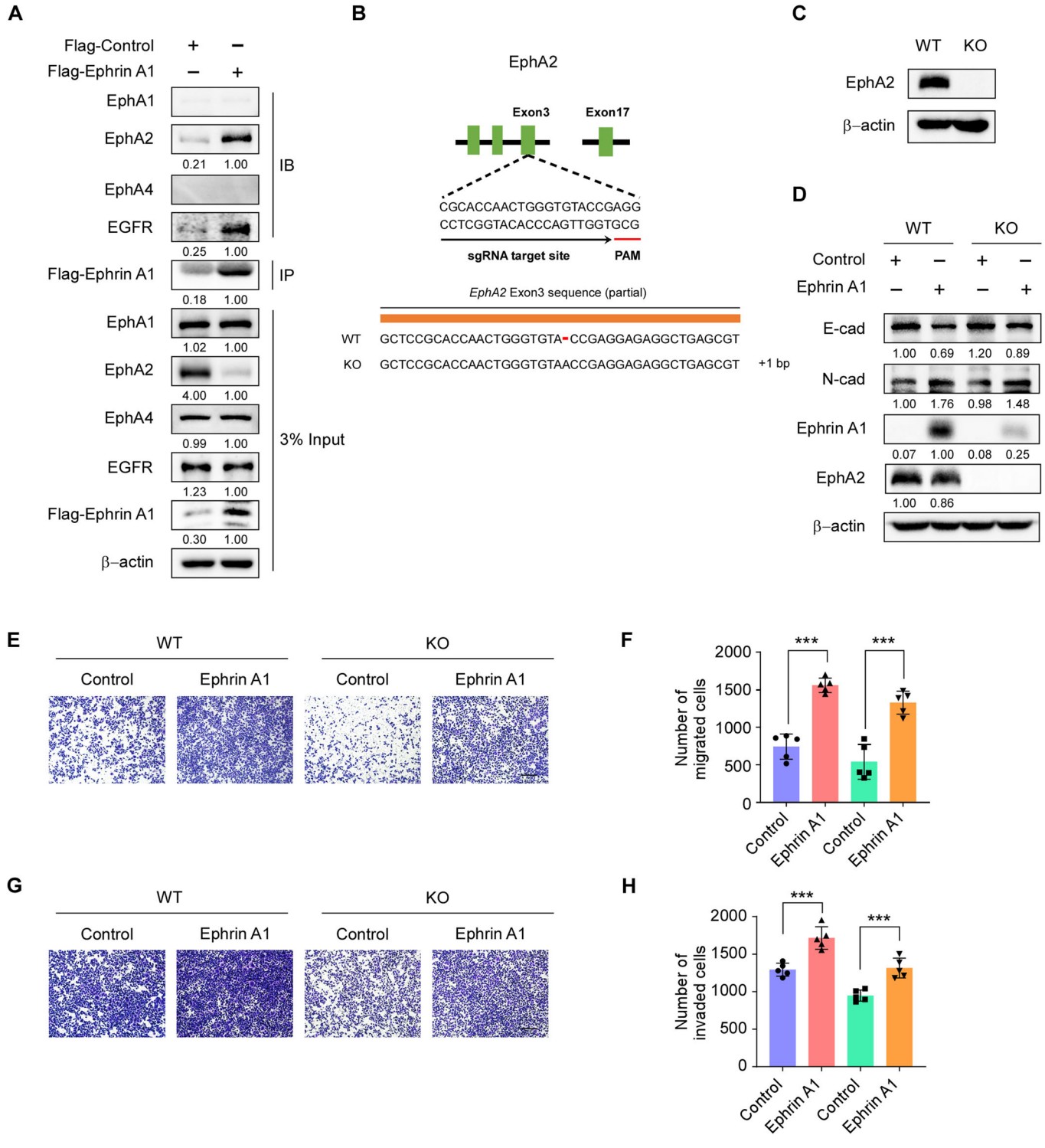

**Figure EV4.  Ephrin A1 inducing EMT in AGS cells is independent of EphA2.**

(A) AGS cells were transfected with control-Flag or Ephrin A1-Flag vectors and applied to immunoprecipitation analysis with anti-Flag beads. The IP samples were subjected to western blot analysis with the indicated antibodies. (B) The construction of *EphA2* knockout (KO) cells by CRISPR-Cas9 system in AGS cells. (C) Western blot analysis of EphA2 expression in AGS wild type (WT) and AGS-*EphA2*-KO cells. (D) Western blot analysis of the expression of EMT markers in control and Ephrin A1-overexpressing AGS cells or *EphA2*-KO AGS cells. (E–H) Transwell migration (E) and invasion (G) analyses of AGS and AGS-*EphA2*-KO cells transfected with control or Ephrin A1 plasmid. The migrated and invaded cells were counted (F, H). *P* values from left to right (F), $P = 5.68\text{e-}6$, $P = 8.97\text{e-}6$. *P* values from left to right (H), $P = 0.0001$, $P = 0.0006$. Scale bars, 200 µm. Experiments were performed three times of biological replicates. Data are shown as mean ± SD. Statistical significance was determined one-way ANOVA. ***$P < 0.001$. Source data are available online for this figure.

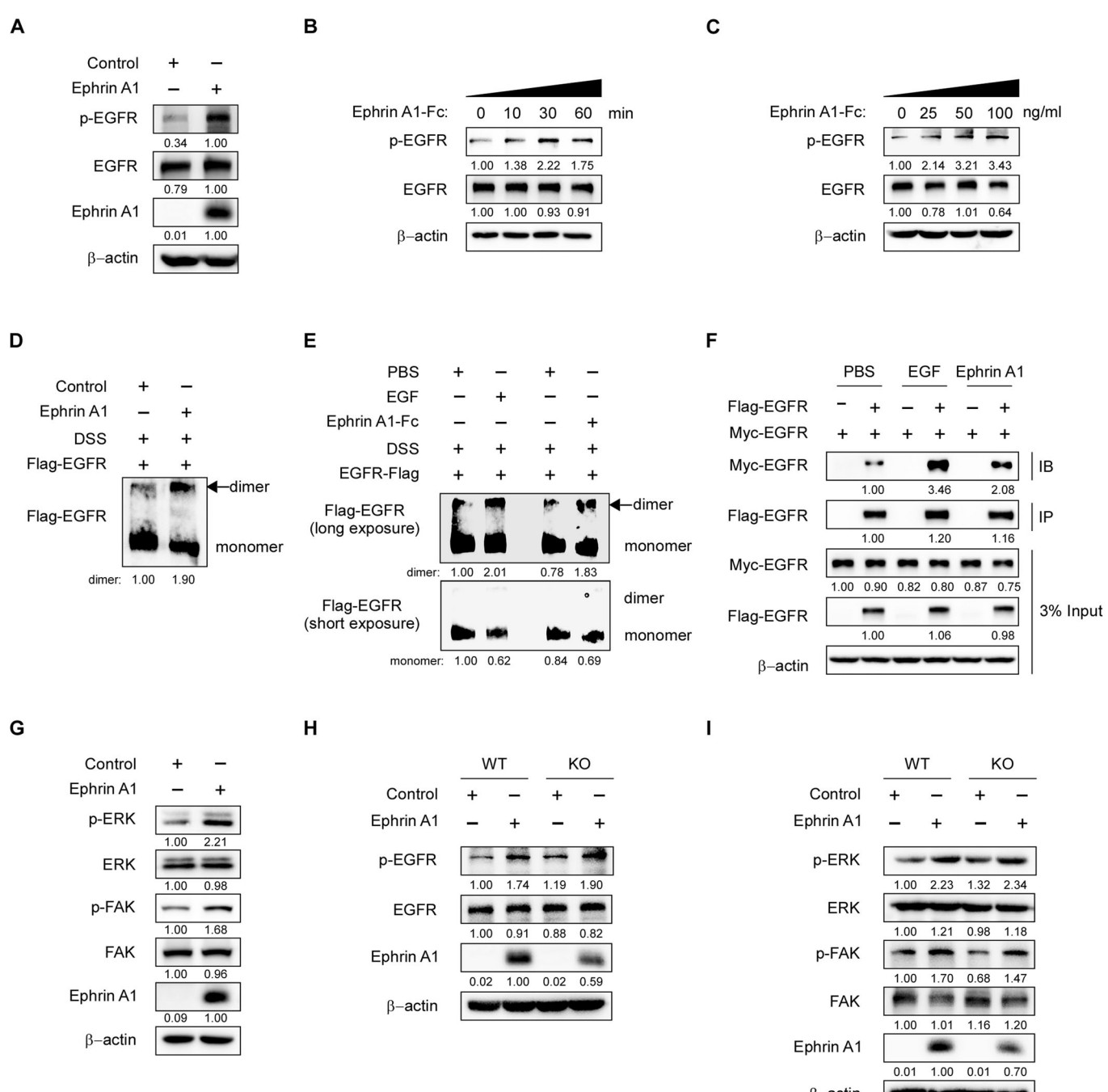

**Figure EV5. Ephrin A1 activates EGFR signaling in AGS cells.**

(A) Western blot analysis of phosphorylation level of EGFR in control or Ephrin A1-overexpressiong AGS cells. (B, C) AGS cells were treated with 100 ng/ml Ephrin A1-Fc for indicated times or treated with different concentrations of Ephrin A1-Fc for 30 min, and then subjected to western blot analysis. (D) Control and Ephrin A1-overexpressing AGS cells were collected and crosslinked with disuccinimidyl suberate (DSS) treatment, followed by western blot analysis. (E) AGS cells were treated with PBS, 100 ng/ml EGF or 1 μg/ml Ephrin A1-Fc proteins and crosslinked with DSS to detect the dimerization level of EGFR. (F) AGS cells were transfected with EGFR-Flag and EGFR-Myc plasmids. After 24 h, the cells were treated with or without PBS, 100 μg/ml EGF or 1 μg/ml Ephrin A1-Fc. The cells were then immunoprecipitated with anti-Flag beads and subjected to western blot analysis. (G) Western blot analysis of EGFR downstream signaling in control and Ephrin A1-overexpressiong AGS cells with indicated antibodies. (H, I) Western blot analysis of the phosphorylation levels of EGFR (H), ERK and FAK (I) in control or Ephrin A1-overexpressiong AGS and AGS-*EphA2*-KO cells. Experiments were performed three times of biological replicates. Source data are available online for this figure.

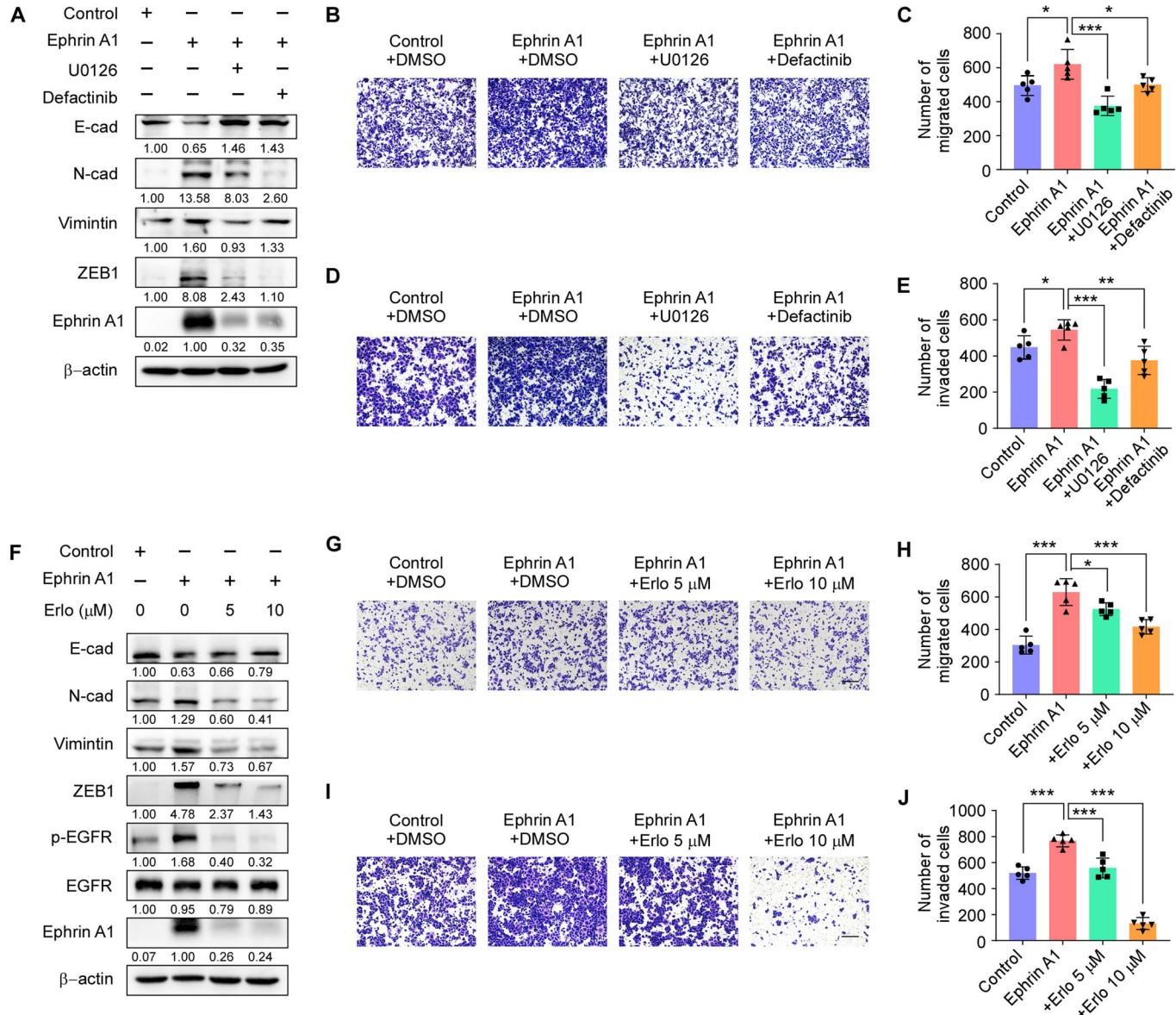

**Figure EV6. Ephrin A1 promotes EMT of gastric cancer cells through EGFR activation.**

(A) Control or Ephrin A1-overexpressing AGS cells were pre-treated with DMSO, 10 μM U0126 or 1 μM defactinib separately and applied to western blot analysis with indicated antibodies. (B–E) The transwell migration (B) and invasion (C) assays of control or Ephrin A1-overexpressing AGS cells that treated with 10 μM U0126 and 1 μM defactinib or not. The migrated and invaded cells were counted (C, E). *P* values from left to right (C), *P* = 0.0306, *P* = 8.10e-5, *P* = 0.0387. *P* values from left to right (E), *P* = 0.0490, *P* = 6.86e-7, *P* = 0.0012. Scale bars, 200 μm. (F) Control or Ephrin A1-overexpressing AGS cells were treated with DMSO or erlotinib and applied to western blot analysis with indicated antibodies. (G–J) The transwell migration (G) and invasion (I) analyses in control or Ephrin A1-overexpressing AGS cells that treated with DMSO or erlotinib at indicated concentration. The migrated and invaded cells were counted (H, J). *P* values from left to right (H), *P* = 7.10e-7, *P* = 0.0486, *P* = 0.0001. *P* values from left to right (J), *P* = 1.39e-5, *P* = 0.0001, *P* = 2.21e-11. Scale bars, 200 μm. Experiments were performed three times of biological replicates. Data are shown as mean ± SD. Statistical significance was determined one-way ANOVA. *$P < 0.05$, **$P < 0.01$, ***$P < 0.001$. Source data are available online for this figure.

