## [Peer Review File · The EMBO Journal]

Ephrin A1 functions as a ligand of EGFR to promote EMT and metastasis in gastric cancer

Shuang Li, Meng Sun, Yun Cui, Dongyang Guo, Feng Yang, Qiang Sun, Yinuo Ding, Mengjie Li, Yiman Liu, Guangshuo Ou, Wei Zhuo, and Tianhua Zhou

Corresponding authors: Wei Zhuo (0012049@zju.edu.cn) , Tianhua Zhou (tzhou@zju.edu.cn)

Review Timeline:

Submission Date:	28th Apr 24
Editorial Decision:	21st Jun 24
Revision Received:	12th Nov 24
Editorial Decision:	5th Dec 24
Revision Received:	16th Dec 24
Accepted:	20th Dec 24

Editor: Daniel Klimmeck

Transaction Report:

Dear Dr Zhuo,

Thank you for submitting your manuscript for consideration by the EMBO Journal, as well as for your patience with our feedback at this time of the year. Your work has now been seen by three referees with expertise in cancer biology and plasticity signaling, whose comments are shown below.

Given the overall interest stated and broader angle of your findings, we are able to invite you to revise your manuscript experimentally to address the referees' comments. I need to stress though that we do require strong support from the referees on a revised version of the study in order to move on to publication of the work.

I would appreciate if you could contact me during the next weeks for exchange e.g. a video call to discuss your perspective on the comments and potential plan for revisions.

Please feel free to contact me if you have any questions or need further input on the referee comments.

When submitting your revised manuscript, please carefully review the instructions below.

Please feel free to approach me any time should you have additional questions related to this.

Thank you for the opportunity to consider your work for publication.

I look forward to your revision.

Best regards,

Daniel Klimmeck

Daniel Klimmeck, PhD
Senior Editor
The EMBO Journal

Instruction for the preparation of your revised manuscript:

- 1) a .docx formatted version of the manuscript text (including legends for main figures, EV figures and tables). Please make sure that the changes are highlighted to be clearly visible.
- 2) individual production quality figure files as .eps, .tif, .jpg (one file per figure).
- 3) a .docx formatted letter INCLUDING the reviewers' reports and your detailed point-by-point response to their comments. As part of the EMBO Press transparent editorial process, the point-by-point response is part of the Review Process File (RPF), which will be published alongside your paper.
- 4) a complete author checklist, which you can download from our author guidelines ([https://wol-prod-cdn.literatumonline.com/pb-assets/embo-site/Author Checklist%20-%20EMBO%20J-1561436015657.xlsx](https://wol-prod-cdn.literatumonline.com/pb-assets/embo-site/Author%20Checklist%20-%20EMBO%20J-1561436015657.xlsx)). Please insert information in the checklist that is also reflected in the manuscript. The completed author checklist will also be part of the RPF.
- 5) Please note that all corresponding authors are required to supply an ORCID ID for their name upon submission of a revised manuscript.
- 6) It is mandatory to include a 'Data Availability' section after the Materials and Methods. Before submitting your revision, primary datasets produced in this study need to be deposited in an appropriate public database, and the accession numbers and database listed under 'Data Availability'. Please remember to provide a reviewer password if the datasets are not yet public (see <https://www.embopress.org/page/journal/14602075/authorguide#datadeposition>). In case you have no data that requires deposition in a public database, please state so in this section. Note that the Data

Availability Section is restricted to new primary data that are part of this study.

7) Our journal encourages inclusion of *data citations in the reference list* to directly cite datasets that were re-used and obtained from public databases. Data citations in the article text are distinct from normal bibliographical citations and should directly link to the database records from which the data can be accessed. In the main text, data citations are formatted as follows: "Data ref: Smith et al, 2001" or "Data ref: NCBI Sequence Read Archive PRJNA342805, 2017". In the Reference list, data citations must be labeled with "[DATASET]". A data reference must provide the database name, accession number/identifiers and a resolvable link to the landing page from which the data can be accessed at the end of the reference. Further instructions are available at .

8) At EMBO Press we ask authors to provide source data for the main and EV figures. Our source data coordinator will contact you to discuss which figure panels we would need source data for and will also provide you with helpful tips on how to upload and organize the files.

Numerical data can be provided as individual .xls or .csv files (including a tab describing the data). For 'blots' or microscopy, uncropped images should be submitted (using a zip archive or a single pdf per main figure if multiple images need to be supplied for one panel). Additional information on source data and instruction on how to label the files are available at .

9) We replaced Supplementary Information with Expanded View (EV) Figures and Tables that are collapsible/expandable online (see examples in <https://www.embopress.org/doi/10.15252/embj.201695874>). A maximum of 5 EV Figures can be typeset. EV Figures should be cited as 'Figure EV1, Figure EV2' etc. in the text and their respective legends should be included in the main text after the legends of regular figures.

11) For data quantification: please specify the name of the statistical test used to generate error bars and P values, the number (n) of independent experiments (specify technical or biological replicates) underlying each data point and the test used to calculate p-values in each figure legend. The figure legends should contain a basic description of n, P and the test applied. Graphs must include a description of the bars and the error bars (s.d., s.e.m.).

We realize that it is difficult to revise to a specific deadline. In the interest of protecting the conceptual advance provided by the work, we recommend a revision within 3 months (19th Sep 2024). Please discuss the revision progress ahead of this time with the editor if you require more time to complete the revisions.

Referee #1:

In this manuscript 'Ephrin A1 functions as a ligand of EGFR to promote EMT and metastasis of gastric cancer cells', the authors investigated the mechanisms of Ephrin A1 in inducing EMT and metastasis. Interestingly, despite that Ephrin A1 interacts with EphA2, a canonical receptor for Ephrin A1, its effects on EMT are independent of EphA2. Subsequently, the authors performed an Immunoprecipitation (IP) screening on other RTKs and found EGFR interacts with EphrinA1. Furthermore, they characterized the domains on EGFR that mediates this interaction. They also carried out a series of experiments such as genetic KO and treatment of EGFR inhibitors to elucidate that EGFR mediates the effects of Ephrin A1. Lastly, the group examined the expression of EphrinA1 and p-EGFR in clinical patient samples. The clinical investigations showed a positive correlation between Ephrin A1 and p-EGFR, supporting their hypothesis that Ephrin A1 activates EGFR downstream signaling. Survival analysis on patients indicates Ephrin A1 as a prognostic marker for gastric cancer patients and highlighted the clinical relevance of this study. This study offered abundant evidence to prove Ephrin A1 as an important factor promoting gastric cancer EMT and metastasis and I suggest accepting this manuscript after addressing the following minor issues:

Major comments:

1. In Supplemental figure 1A, the authors showed the expression of Ephrin A1 across different gastric cancer cell lines. I am curious whether the intrinsic properties of those cell lines, such as EMT status, metastatic abilities, colony formation abilities etc. are correlated with their Ephrin A1 expression status or not. This information may help to further support their hypothesis of Ephrin A1 in driving EMT and metastasis.
2. The authors elucidated the domains of EGFR that are required for interacting with Ephrin A1. Although the signal peptide was preserved, it is still important to examine the localization of these truncated proteins. This will determine whether the loss of binding is due to the absence of specific domains or the overall loss of localization on the cell membrane. It may also be helpful to express these truncated EGFRs in cells and perform some functional assays to further elucidate the importance of Ephrin A1 binding to EGFR.
3. AGS, MKN45 and NCI-N87 cells seem to be used interchangeably. Specifically, NCI-N87 was used in multiple functional assays and in vivo models; AGS and MKN45 were primarily used in investigating the interactions between Ephrin A1 and EGFR. I have some concerns about whether they show similar phenotypes in vivo. Especially for MKN45 that expresses relatively high levels of Ephrin A1, does it metastasize more than NCI-N87 cells at baseline level? Does Ephrin A1 KD and/or KO on MKN45 decrease lung metastasis?
4. In page 13, one very simplified sentence was used to describe Figure 7D. The figure legend for this panel was also vague. Please elaborate in the text on how this data showed Ephrin A1 as an independent prognostic factor in gastric cancer patients.
5. In the proposed model shown in Figure 7F, the Ephrin A1 binding to EGFR and its downstream activation happen in the primary tumor, but there was no data showing their effects in driving primary tumor growth or spontaneous metastasis. More data on these aspects are needed for this current schematic model, otherwise this zoom-in view should be in the cells in the blood stream, which is in line with this study using tail vein injection for a metastasis model. Please make this adjustment accordingly.

Minor comments:

1. The authors often refer to many panels at one time. For example, in page 12, Figures 4B-I and figures 5D-K were described only by one sentence, respectively. It may be better to elaborate on these panels or at least mention these panels closely following descriptions for each experiment. i.e. you may change this sentence to 'Furthermore, we evaluated the cell migration (Figs. 5D-E), invasion (Figs. 5F-G), 3D colony formation (Figs. 5H-I), and transendothelial cell migration abilities (Figs. 5J-K) of gastric cancer cells...' so that we readers can follow the text better.
2. For statistical analysis on more than two groups, One-Way ANOVA with multiple comparisons is usually preferred over student t test.
3. Please be consistent with the data presentation format or at least give the accurate annotations in figure legends. In Figures 4E, 4I and 5G, there were no lower error bars, although it was stated as 'data are shown as mean SD'.

Other non-essential suggestions

1. In the discussion part, the authors mentioned that a co-crystal structure between Ephrin A1 and EGFR worth further exploration. A bit elaboration on why this is essential may help support this claim. For example, maybe knowing the structural basis of their binding will help to explain why the domains characterized in this study are required. This will also potentially facilitate the development of small molecules specifically targeting this binding.
2. The authors may want to refine the text further to make sure there are no grammar errors, and the flow of the text is smooth. I found a few as listed here:
 - Page 6: 'we used a lentivirus to stably overexpress....' Maybe the meaning was 'we used a lentiviral system to...'
 - Page 7: 'Since cancer cells undergone EMT can acquire invasive phenotypes to initiate metastatic dissemination. Therefore, we evaluated...' this should be in the same sentence and there should not have a period before 'Therefore'.
 - Page 12: 'Together, these results suggest that Ephrin A1 activates EMT of gastric cancer cells may through activation of ...' 'may' should be deleted or placed right before 'activates'.

Referee #2:

The findings and conclusions presented in this study hold significant interest for researchers in the field.

The author, Shuang Li and colleagues assert that Ephrin A1 is a novel ligand for EGFR, promoting epithelial-mesenchymal transition (EMT) and metastatic dissemination. The article presents substantial data, convincingly demonstrating the role of Ephrin A1 in driving EMT and metastasis.

However, there are critical concerns that need to be addressed to fully support the conclusion that all these phenotypes are the result of EGFR activation by Ephrin A1. Specifically, all experiments in Figure 1 must be conducted with the appropriate control, which in this case is represented by the Eph receptors knockout (KO), as attempted in Figure 2.

Additionally, Figure 3 must be repeated with the proper control (Eph receptors KO), and the same control should be applied to Figures 4 and 6.

Referee #3:

In the manuscript, "Ephrin A1 functions as a ligand of EGFR to promote EMT and metastasis of gastric cancer cells" Li & Cui et al. present data that Ephrin A1 binds EGFR and that Ephrin A1 induces EMT and promotes metastatic behavior. The story is potentially interesting; however, the authors constantly jump between different cell lines telling one part of a story with a certain cell line and then assuming that the exact same thing is happening in a different cell line when they show a different part of the story. The cells they chose have drastically different phenotypes and very different expression of the relevant proteins. The authors need to perform the complete set of experiments (at least ex vivo experiments) on all the cell lines NCI-N87, MKN45, and AGS cells and show that the results are consistent with one another.

For example, they show that EGFR is phosphorylated and dimerized in AGS cells. But then show that downstream EGFR signaling is unregulated in MKN45 cells. The authors need to demonstrate that EGFR is phosphorylated and dimerized in MKN45 cells. They need to show that downstream signaling is upregulated in AGS cells. They should probably do the same with NCI-N87 cells as all the mouse work was done with those cells. And also they should demonstrate that this is independent of EphA2 in MKN45 cells. And independent of what ever EphAx, Ephrin A1 is binding in NCI-N87 cells and AGS cells if they are to argue that the direct interaction between Ephrin A1 and EGFR is responsible for the EMT induced in these cells.

Major comments:

1) Figure 1B and 1D seem discordant. In figure 1B, the western blot suggests that E-cadherin expression is completely lost, while in figure 1D, it shows that E-cadherin has been internalized. As western blot is more sensitive than IF, can the authors reconcile this?

2) The complete absence of forming colonies in soft agar and lung metastases with NCI-N87 cells seems discordant with the literature. Can the authors please reconcile? Were the protocols different in some way?

Several groups have reported NCI-N87 cells from colonies in soft agar. For example Supplemental Figure 3DE from Wang et al. (2020) Gut 69: 1193-1205.

Likewise, others have reported that NCI-N87 cells implant in the lung after tail vein injection in SCID mice, yet this was not observed here. e.g 1/4 as many NCI-N87 cells (5×10^5) yielded lung implants in Zhang (2023) PLoS Genetics 19(2): e1010640 and resulting in death starting at 10 days - Fig 2 J,K

3) The wording from the middle paragraph on pg 9 is not correct "To further ... receptor independent" is wrong. The authors switched to MKN45 cells and as such were not inducing EMT anymore with Ephrin A1 O/E as they did with NCI-N87 cells, they are suppressing an innate tendency for EMT in MKN45 cells with Ephrin A1 knockdown. Induction of EMT needs to have been done with O/E of Ephrin A1 in NCI-N87 cells.

4) Why did Ephrin A1 expression dramatically decreased with knockout of EphA2 (Fig. 2C). Isn't this going to be partially responsible for the phenotype? If EphA2 is knocked out in NCI-N87 cells are they resistant to EMT induction?

5) EphA2 has been shown to interact with EGFR [Kim (2022) Cell death and disease 13:528]. As such the authors should perform the Ephrin A1 pulldown in the MKN45 EphA2 knockout cells to demonstrate that they are not simply pulling down the ternary complex of Ephrin A1-EphA2-EGFR.

6) Why did the authors switch to MKN45 cells in Figure 2? The NCI-N87 EMT experiments seemed much more dramatic.

7) Did the EGFR mutants localize to the apical membrane like wild-type? Cellular mislocalization might explain the lack of binding. Anti-Myc IF on these cells should be performed.

8) Why did the authors then switch to AGS cell lines in figure 3? The authors should repeat all experiments in Figure 2 with the AGS cells to argue that the phosphorylation changes are due to Ephrin A1-EGFR. What EphA2, A3, A4 are in AGS cells. Do

these bind Ephrin? What do experiments with knockout of EphA2 (and A3, A4 if necessary) look like in AGS cells.

9) Which cell line is in figure 3H? - Figure legend says gastric cancer cells and the authors keep on switching between three different lines so it is unclear what cell line this is.

10) Does EGFR get phosphorylated and dimerize in MKN45 cells? This is important because the authors switched back to MKN45 cells at the end of figure 3 (I and J) and said there is downstream signaling from EGFR. But only showed phosphorylation and dimerization in AGS cells in response to Ephrin A1.

11) Figure 4. The authors demonstrate that EMT is blocked with the drugs in AGS cells, but then test the soft agar and TEM in NCI-N87 cells. Please demonstrate that EMT is blocked in the NCI-N87 cells with these drugs when Ephrin is added.

12) Figure 5. Quantitation of the western blots (throughout) would be useful. Looks like ZEB1 and N-Cad decrease in the EGFR KO (panel B). Does EGFR knockout in NCI-N87 cells block EMT with Ephrin A1.

13) Figure 6. The authors have not demonstrated that EGFR signaling (phosphorylation, dimerization, and downstream targets) is altered in NCI-N87 cells. These were shown in AGS and MKN45 cells.

14) Can the authors demonstrate that inhibition of EphA2 does not block the Ephrin phenotype. Some inhibitors listed here: Xiao (2020) Journal of hematology and oncology 13:114. Dasatinib: Chang et al (2008) Br J Cancer 99(7): 1074.

Minor:

-Re: Fig 3A. Ideally, the reverse IP would be done with EGFR (lacking the Fc region)

https://www.rndsystems.com/products/recombinant-human-egfr-protein_1095-er. It is not very expensive at 150 dollars (1095-ER)

-I do NOT doubt that knockouts were made based on the western blots presented; however, CRISPR makes random indels so it would be rare to have the single indel on both chromosome (or all if they are hyper diploid or tetraploid) as suggested by the figures.

Point-by-point response to reviewer comments :**Reply to Referee #1:**

In this manuscript 'Ephrin A1 functions as a ligand of EGFR to promote EMT and metastasis of gastric cancer cells', the authors investigated the mechanisms of Ephrin A1 in inducing EMT and metastasis. Interestingly, despite that Ephrin A1 interacts with EphA2, a canonical receptor for Ephrin A1, its effects on EMT are independent of EphA2. Subsequently, the authors performed an Immunoprecipitation (IP) screening on other RTKs and found EGFR interacts with EphrinA1. Furthermore, they characterized the domains on EGFR that mediates this interaction. They also carried out a series of experiments such as genetic KO and treatment of EGFR inhibitors to elucidate that EGFR mediates the effects of Ephrin A1. Lastly, the group examined the expression of EphrinA1 and p-EGFR in clinical patient samples. The clinical investigations showed a positive correlation between Ephrin A1 and p-EGFR, supporting their hypothesis that Ephrin A1 activates EGFR downstream signaling. Survival analysis on patients indicates Ephrin A1 as a prognostic marker for gastric cancer patients and highlighted the clinical relevance of this study. This study offered abundant evidence to prove Ephrin A1 as an important factor promoting gastric cancer EMT and metastasis and I suggest accepting this manuscript after addressing the following minor issues:

We really appreciate the reviewer's positive and insightful comments regarding our works.

Major comments:

1. In Supplemental figure 1A, the authors showed the expression of Ephrin A1 across different gastric cancer cell lines. I am curious whether the intrinsic properties of those cell lines, such as EMT status, metastatic abilities, colony formation abilities etc. are correlated with their Ephrin A1 expression status or not. This information may help to further support their hypothesis of Ephrin A1 in driving EMT and metastasis.

Thank you very much for your constructive comments. According to the reviewer's good suggestion, we detected the EMT status, migration, invasion and colony formation abilities of different gastric cancer cell lines. The results showed that Ephrin A1 low expression cells (NCI-N87, AGS and HGC-27 cells) had low expression level of stromal cell marker N-cadherin. While Ephrin A1 high expression cells (MKN45, BGC-823 and SGC-7901 cells) had high expression level of N-cadherin. It suggested that the expression level of Ephrin A1 may be correlated with the EMT status of cells (**Fig. R1A**). Therefore, in our study, NCI-N87 and AGS cells were used for Ephrin A1 exogenous expression and EMT induction experiments. MKN45 cells were used for Ephrin A1 knock-down and EMT inhibition assays.

In addition, the migration (**Fig. R1B**), invasion (**Fig. R1C**) and colony formation (**Fig. R1D**) abilities of NCI-N87 cells were very weak, which was consistent with the low expression of Ephrin A1 (**Figs. R1B-D**). MKN45 cells had high expression level of Ephrin A1 and the migration, invasion and colony formation abilities were relatively strong (**Figs. R1B-D**).

However, there was no consistent correlation in all cell lines between Ephrin A1 expression level and the ability of metastasis and clone formation (**Figs. R1B-D**). Because tumor metastasis is a multi-step process with complex molecular changes (Chaffer et al., 2011; Shi et al., 2024), it is understandable that the expression of one gene cannot determine the metastasis ability among different cells.

Figure for reviewers removed

References:

Chaffer CL, Weinberg RA. A perspective on cancer cell metastasis. *Science*. 2011;331(6024):1559-1564.

Shi X, Wang X, Yao W, et al. Mechanism insights and therapeutic intervention of tumor metastasis: latest developments and perspectives. *Signal Transduct Target Ther*. 2024;9(1):192.

2. The authors elucidated the domains of EGFR that are required for interacting with Ephrin A1. Although the signal peptide was preserved, it is still important to examine the localization of these truncated proteins. This will determine whether the loss of binding is due to the absence of specific domains or the overall loss of localization on the cell membrane. It may also be helpful to express these truncated EGFRs in cells and perform some functional assays to further elucidate the importance of Ephrin A1 binding to EGFR.

It is really a good suggestion. According to your kind suggestion, we expressed EGFR and truncated EGFRs in cells and detected the localization of these proteins by immunofluorescence with anti-Myc antibody. The results showed that truncated EGFR (Δ ECD, Δ D1, Δ D3, Δ D1+D3) proteins distributed in both cell membrane and cytoplasm and had similar localization pattern with full-length EGFR proteins (**Fig. R2**).

Figure for reviewers removed

In addition, we expressed EGFR and the truncated EGFRs in Ephrin A1-overexpressing AGS cells (**Fig. R3A**) and detected cell migration and invasion abilities. Our data showed that compared with wild type EGFR, the number of migrated (**Figs. R3B, D**) and invaded (**Figs. R3C, E**) cells were significantly reduced in truncated EGFR cells, which suggested that domains I and III of EGFR ECD were required for the function of Ephrin A1.

Figure for reviewers removed

3. AGS, MKN45 and NCI-N87 cells seem to be used interchangeably. Specifically, NCI-N87 was used in multiple functional assays and in vivo models; AGS and MKN45 were primarily used in investigating the interactions between Ephrin A1 and EGFR. I have some concerns about whether they show similar phenotypes in vivo. Especially for MKN45 that expresses relatively high levels of Ephrin A1, does it metastasize more than NCI-N87 cells at baseline level? Does Ephrin A1 KD and/or KO on MKN45 decrease lung metastasis?

We really appreciate your valuable advice. As you suggested, we constructed MKN45-Ephrin A1 KD cells and then assessed the lung metastasis abilities of luciferase-labeled NCI-N87, MKN45 and MKN45-Ephrin A1 KD cells by lung metastasis mouse model (**Figs. R4A-B**). The bioluminescence imaging (**Figs. R4C-D**) and H&E staining (**Figs. R4E-F**) of lungs showed that MKN45 cells metastasized more than NCI-N87 cells at baseline level. Furthermore, Ephrin A1 KD on MKN45 cells significantly decreased lung metastasis compared with control cells (**Figs. R4C-F**), which further confirmed that Ephrin A1 promoted the metastasis ability of gastric cancer cells.

Considering the significance of these results, we have included **Fig. R4** as **new Figs. EV3G-L** in our revised manuscript, and the corresponding description was also included in page 8 of our current manuscript.

Figure for reviewers removed

4. In page 13, one very simplified sentence was used to describe Figure 7D. The figure legend for this panel was also vague. Please elaborate in the text on how this data showed Ephrin A1 as an independent prognostic factor in gastric cancer patients.

Thanks a lot for this good concern. We have modified our expression as below and described Fig. 7D in detail in page 14 of our revised manuscript.

Fig. 7D: Further multivariate Cox analysis showed that high expression level of Ephrin A1 was an independent positive prognostic factor for predicting outcomes of gastric cancer patients (HR: 2.26; $P < 0.05$).

5. In the proposed model shown in Figure 7F, the Ephrin A1 binding to EGFR and its downstream activation happen in the primary tumor, but there was no data showing their effects in driving primary tumor growth or spontaneous metastasis. More data on these aspects are needed for this current schematic model, otherwise this zoom-in view should be in the cells in the blood stream, which is in line with this study using tail vein injection for a metastasis model. Please make this adjustment accordingly.

Thank you very much for the constructive suggestion. We have modified the zoom-in view from the tumor cells in the blood stream. Our new schematic model was as below (**Fig. R5**). According to the reviewer's good suggestion, we have updated the model as the **new Fig. 7F** in our revised manuscript.

Figure for reviewers removed

Minor comments:

1. The authors often refer to many panels at one time. For example, in page 12, Figures 4B-I and figures 5D-K were described only by one sentence,

respectively. It may be better to elaborate on these panels or at least mention these panels closely following descriptions for each experiment. i.e. you may change this sentence to 'Furthermore, we evaluated the cell migration (Figs. 5D-E), invasion (Figs. 5F-G), 3D colony formation (Figs. 5H-I), and transendothelial cell migration abilities (Figs. 5J-K) of gastric cancer cells...' so that we readers can follow the text better.

Thanks a lot for this good concern. We have modified our descriptions of the figures as below and updated in our revised manuscript.

① Figs. EV2A-H: We found that ectopic expression of Ephrin A1 significantly promoted migration and invasion abilities of NCI-N87 (Figs. EV2A-D) and AGS cells (Figs. EV2E-H). The corresponding description was in page 7 of our revised manuscript.

② Figs. 1N-R: Ephrin A1 significantly promoted the lung metastasis of gastric cancer cells (Figs. 1N-Q) and reduced overall survival time of mice (Fig. 1R). The corresponding description was in page 8 of our revised manuscript.

③ Figs. 4B-I: The results showed that U0126 or defactinib treatment significantly suppressed the EMT process (Fig. 5C) and inhibited the 3D colony formation (Figs. 5D, E) and transendothelial migration (Figs. 5F, G) abilities induced by Ephrin A1 in NCI-N87 cells. Moreover, blockage of ERK and FAK activation also inhibited EMT (Fig. EV6A), migration (Figs. EV6B, C) and invasion (Figs. EV6D, E) abilities induced by Ephrin A1 in AGS cells.

The Figs. 4B-I were undated as new Figs. 5D-G and Figs. EV6B-E, and the corresponding descriptions were in page 13 of our revised manuscript.

④ Figs. 5D-K: We evaluated the 3D colony formation and transendothelial migration abilities of NCI-N87 cells and found that erlotinib treatment significantly inhibited the functions of Ephrin A1 (Figs. 5I-L). In addition, the EMT (Fig. EV6F), migration (Fig. EV6G, H) and invasion (Fig. EV6I, J) abilities induced by Ephrin A1 were significantly suppressed in AGS cells treatment with erlotinib.

The Figs. 5D-K were undated as new Figs. 5I-L and Figs. EV6G-J, and the corresponding description was in page 13 of our revised manuscript.

2. For statistical analysis on more than two groups, One-Way ANOVA with multiple comparisons is usually preferred over student t test.

Thanks very much for your valuable suggestion. We have reanalyzed the statistical data of more than two groups by one-way ANOVA in our revised manuscript.

3. Please be consistent with the data presentation format or at least give the accurate annotations in figure legends. In Figures 4E, 4I and 5G, there were no lower error bars, although it was stated as 'data are shown as mean \pm SD'.

We apologize for not making consistent with the data presentation format and thank you for highlighting this point. We have updated the Figures (4E, 4I and 5G) and the lower error bars were shown (**Fig. R6**). In our current manuscript, we have updated them as **new Fig. EV6E, Fig. 5G and Fig. EV6J**.

Figure for reviewers removed

Other non-essential suggestions

1. In the discussion part, the authors mentioned that a co-crystal structure between Ephrin A1 and EGFR worth further exploration. A bit elaboration on why this is essential may help support this claim. For example, maybe knowing the structural basis of their binding will help to explain why the domains characterized in this study are required. This will also potentially facilitate the development of small molecules specifically targeting this binding.

We really appreciate your insightful suggestion. We have discussed this point as following and added the description in the Discussion section of our current manuscript (page 16).

Discussion: The co-crystal structure between Ephrin A1 and EGFR interaction is important and worthwhile to be further explored. It may help to explain why the domains characterized in this study are required and potentially facilitate the development of small molecules specifically targeting this binding.

2. The authors may want to refine the text further to make sure there are no grammar errors, and the flow of the text is smooth. I found a few as listed here:

- Page 6: 'we used a lentivirus to stably overexpress....' Maybe the meaning was 'we used a lentiviral system to...'

- Page 7: 'Since cancer cells undergone EMT can acquire invasive phenotypes to initiate metastatic dissemination. Therefore, we evaluated...' this should be in the same sentence and there should not have a period before 'Therefore'.
- Page 12: 'Together, these results suggest that Ephrin A1 activates EMT of gastric cancer cells may through activation of ...' 'may' should be deleted or placed right before 'activates'.

Thanks very much for your attention to detail, which help us improve our manuscript. We polished our manuscript carefully and the sentences pointed out were modified as below.

Page 6: We stably overexpressed human Ephrin A1 in NCI-N87 cells by the lentiviral system. The corresponding description was updated in page 6 of our revised manuscript.

Page 7: Since cancer cells undergone EMT can acquire invasive phenotypes to initiate metastatic dissemination, therefore we evaluated whether Ephrin A1 induces the invasive phenotypes of gastric cancer cells. The corresponding description was updated in page 7 of our revised manuscript.

Page 12: Together, these results suggest that Ephrin A1 may induce EMT of gastric cancer cells through activating EGFR signaling. The corresponding description was updated in page 13 of our revised manuscript.

Reply to Referee #2:

The findings and conclusions presented in this study hold significant interest for researchers in the field.

The author, Shuang Li and colleagues assert that Ephrin A1 is a novel ligand for EGFR, promoting epithelial-mesenchymal transition (EMT) and metastatic dissemination. The article presents substantial data, convincingly demonstrating the role of Ephrin A1 in driving EMT and metastasis.

However, there are critical concerns that need to be addressed to fully support the conclusion that all these phenotypes are the result of EGFR activation by Ephrin A1. Specifically, all experiments in Figure 1 must be conducted with the appropriate control, which in this case is represented by the Eph receptors knockout (KO), as attempted in Figure 2.

Additionally, Figure 3 must be repeated with the proper control (Eph receptors KO), and the same control should be applied to Figures 4 and 6.

Thank you very much for your constructive comments on our previous manuscript, which are crucial for improving the credibility and accuracy of our manuscript.

To explore the underlying mechanism of how Ephrin A1 induces EMT and metastasis of gastric cancer cells, we employed immunoprecipitation assay to identify the interactions of Ephrin A1 with the reported Eph receptors. Our results showed that Ephrin A1 interacted with EphA2. Furthermore, we constructed *EphA2* knockout cell line in MKN45 cells, and showed that the function of Ephrin A1 in regulating EMT of gastric cancer cells is EphA2 independent (**Figs. 2B-D**). Therefore, it is appropriate to use *EphA2* KO as control to repeat the phenotypes. Because MKN45 cells have high expression level of Ephrin A1, we used to perform Ephrin A1-knockdown assays in the study.

(1) Here, we use MKN45-*EphA2*-KO cells to repeat the Ephrin A1-knockdown phenotypes:

Firstly, we detected the EMT status in *EphA2* KO cells and found that downregulation of Ephrin A1 significantly inhibited EMT process in MKN45-*EphA2*-KO cells (**Fig. R7A**). Furthermore, to determine whether knockdown of Ephrin A1 could inhibit metastasis abilities of *EphA2* KO cells, we performed transwell migration, invasion and 3D soft agar colony formation assays. The results showed that knockdown of Ephrin A1 expression markedly suppressed migration (**Figs. R7B-C**), invasion (**Figs. R7D-E**) and colony formation (**Figs. R7F-G**) abilities of MKN45-*EphA2*-KO cells. These data were consistent with the results in MKN45 wild type cells (**Figs. EV2I-L and EV3A-B**), indicating

that the function of Ephrin A1 in regulating EMT and metastasis is EphA2 receptor independent. According to the reviewer's good suggestion, we have added **Figs. R7B-G** as **new Figs. 2E-J**, and the corresponding description was included in Page 9 of our current manuscript.

Figure for reviewers removed

In addition, we performed lung metastasis assay to assess the function of Ephrin A1 in the absent of EphA2 *in vivo* (**Fig. R8A**). The bioluminescence imaging showed that downregulation of Ephrin A1 in MKN45-*EphA2*-KO cells significantly decreased lung metastasis compared with control cells (**Figs. R8B-C**), which were consistent with the results in MKN45 wild type cells (**Fig. R4**). Considering the significance of these results, we have included **Fig. R8** as **new Figs. 2K-M** in our revised manuscript, and the corresponding description was also included in Pages 10.

Figure for reviewers removed

Moreover, to verify the functional role of Ephrin A1 in EGFR activation, we employed immunoprecipitation assay and found that Ephrin A1 interacted with EGFR in the absent of EphA2 (**Fig. R9A**). Then, we evaluated the activation of EGFR by Ephrin A1. Western blot assay showed that knockdown of Ephrin A1 significantly inhibited the phosphorylation level of EGFR (**Fig. R9B**) and suppressed the activation of EGFR downstream signaling (**Fig. R9C**) in MKN45-*EphA2*-KO cells. In our current manuscript, we have added **Fig. R9** as **new Fig.3C** and **Figs. 4J-K**, and the corresponding description was also included in Pages 10 and 12.

Figure for reviewers removed

(2) To further demonstrate that the function of Ephrin A1 in inducing EMT and metastasis of gastric cancer cells is EphA2 independent, we firstly detected the receptors binding to Ephrin A1 in AGS cells and found that Ephrin A1 interacted with EphA2 and EGFR (**Fig. R10A**), but not EphA1 and EphA4, which was consistent with the results in Fig2. Therefore, we generated *EphA2* knockout cell line in AGS cells (**Fig. R10B**), which had low expression level of Ephrin A1 and were used for Ephrin A1 exogenous expression assays (**Fig. EV1A**). Western blot and DNA sequencing data showed that the KO cells were established successfully (**Figs. R10C-D**).

Then we ectopically expressed Ephrin A1 and found that Ephrin A1 significantly promoted EMT process both in wild type AGS cells and AGS-*EphA2*-KO cells (**Fig. R10E**). In addition, the results of transwell migration and invasion assays showed that ectopic expression of Ephrin A1 markedly facilitated migration (**Figs. R10F-G**) and invasion (**Figs. R10H-I**) abilities of AGS-*EphA2*-KO cells, which were consistent with the results in AGS cells. These data demonstrated that Ephrin A1 could promote EMT and metastasis abilities in AGS-*EphA2*-KO cells. Considering the significance of these results, we have included **Fig. R10** as **new Fig. EV4** in our revised manuscript, and the corresponding description was in pages 9-10.

Figure for reviewers removed

Furthermore, we evaluated whether Ephrin A1 could activate EGFR signaling. The results showed that ectopic expression of Ephrin A1 significantly increased the phosphorylation level of EGFR (**Fig. R11A**) and promoted the activation of EGFR downstream signaling (**Fig. R11B**) both in AGS and AGS-*EphA2*-KO cells. These results suggested that Ephrin A1 may induce EMT of gastric cancer cells through activation of EGFR signaling. Here, we have added **Fig. R11** as **new Figs. EV5H-I** in our current manuscript, and the corresponding description was included in Page 12.

Figure for reviewers removed

In summary, our findings in MKN45-*EphA2*-KO cells and AGS-*EphA2*-KO cells indicated that Ephrin A1 regulated EMT and metastasis of gastric cancer cells through EGFR signaling, which was EphA2 receptor independent.

Reply to Referee #3:

In the manuscript, "Ephrin A1 functions as a ligand of EGFR to promote EMT and metastasis of gastric cancer cells" Li & Cui et al. present data that Ephrin A1 binds EGFR and that Ephrin A1 induces EMT and promotes metastatic behavior. The story is potentially interesting; however, the authors constantly jump between different cell lines telling one part of a story with a certain cell line and then assuming that the exact same thing is happening in a different cell line when they show a different part of the story. The cells they chose have drastically different phenotypes and very different expression of the relevant proteins. The authors need to perform the complete set of experiments (at least ex vivo experiments) on all the cell lines NCI-N87, MKN45, and AGS cells and show that the results are consistent with one another.

For example, they show that EGFR is phosphorylated and dimerized in AGS cells. But then show that downstream EGFR signaling is unregulated in MKN45 cells. The authors need to demonstrate that EGFR is phosphorylated and dimerized in MKN45 cells. They need to show that downstream signaling is upregulated in AGS cells. They should probably do the same with NCI-N87 cells as all the mouse work was done with those cells. And also they should demonstrate that this is independent of EphA2 in MKN45 cells. And independent of what ever EphAx, Ephrin A1 is binding in NCI-N87 cells and AGS cells if they are to argue that the direct interaction between Ephrin A1 and EGFR is responsible for the EMT induced in these cells.

Thank the reviewer so much for the important suggestion, which helps us to improve our manuscript. We apologize for making confusing of cell lines used in our manuscript and thank you for highlighting this point. In our study, AGS and NCI-N87 cells have low expression levels of Ephrin A1 and were used for Ephrin A1 exogenous expression and EMT induction assay. Whereas MKN45 cells have high expression level of Ephrin A1 and used for Ephrin A1 knock-down and EMT inhibition experiments (**Fig. EV1A and Fig. R12**).

Figure for reviewers removed

According to the reviewer's suggestion, we completed experiments of EGFR interaction, phosphorylation, dimerization and downstream signaling pathway activation in NCI-N87, MKN45 and AGS cells. The data were shown in the responses below (**Figs. R19, R21, R24, R27**). Furthermore, to reconcile the cell lines used in the study, we rearranged main data from NCI-N87 and MKN45 cells in Figures and data from AGS cells in EV Figures in our revised manuscript.

Major comments:

1) Figure 1B and 1D seem discordant. In figure 1B, the western blot suggests that E-cadherin expression is completely lost, while in figure 1D, it shows that E-cadherin has been internalized. As western blot is more sensitive than IF, can the authors reconcile this?

Thanks a lot for the valuable concern. Both IF and western blot assays showed that overexpression of Ephrin A1 significantly inhibited the expression of E-cadherin (**Figs. R13A, B**). According to the reviewer's suggestion, we adjusted the immunofluorescence results of E-cadherin synchronously (**Fig. R13B**) and added **Fig. R13B** as **new Fig. 1C** in our revised manuscript.

Figure for reviewers removed

2) The complete absence of forming colonies in soft agar and lung metastases with NCI-N87 cells seems discordant with the literature. Can the authors please reconcile? Were the protocols different in some way?

Several groups have reported NCI-N87 cells from colonies in soft agar. For example Supplemental Figure 3DE from Wang et al. (2020) Gut 69:

1193-1205.

Thanks a lot for the reviewer's valuable concern. For 3D soft agar colony formation assay, the magnification of the pictures was different with the literature. The scale bar of Supplemental Figure 3D from Wang et al. (2020) Gut 69: 1193-1205 was 200 μm , and the scale bar in our manuscript Fig. 1F was 400 μm . In our statistical data Fig1. G, only the large visible clones at the current magnification were counted. In addition, we can also see small clones forming in the zoom in view of NCI-N87 cells (**Fig. R14**). Furthermore, the reagents used in different labs may also contributing to the differences of colony formation abilities. Thanks for your understanding in advance.

Figure for reviewers removed

Likewise, others have reported that NCI-N87 cells implant in the lung after tail vein injection in SCID mice, yet this was not observed here. e.g 1/4 as many NCI-N87 cells (5×10^5) yielded lung implants in Zhang (2023) PLoS Genetics 19(2): e1010640 and resulting in death starting at 10 days - Fig 2 J,K

For lung metastasis assay of NCI-N87 cells, the mice used were different. In our manuscript Fig 1N, we used SCID mice, while NOD-SCID mice were used in the Fig 2J in Zhang (2023) PLoS Genetics 19(2): e1010640. According to the literature, the SCID immunodeficient mice is lack of mature T and B cells of the adaptive immune system, but have an intact innate immune system including NK cells and macrophages. However, NOD-SCID mice has simultaneously defective adaptive and innate immunity, resulting in a higher degree of immune deficiency than SCID mice (Shultz et al., 2014; Chen et al., 2022). Therefore, NOD-SCID mice exhibit an increased ability to support engraftment with solid human cancers than SCID mice.

References:

Wang Q, Chen C, Ding Q, et al. METTL3-mediated m6A modification of HDGF mRNA promotes gastric cancer progression and has prognostic significance. *Gut*. 2020;69(7):1193-1205.

Zhang FF, Jiang C, Jiang DP, et al. ARHGAP15 promotes metastatic colonization in gastric cancer by suppressing RAC1-ROS pathway. *PLoS Genet*. 2023;19(2):e1010640.

Shultz LD, Goodwin N, Ishikawa F, Hosur V, Lyons BL, Greiner DL. Human cancer growth and therapy in immunodeficient mouse models. *Cold Spring Harb Protoc*. 2014;2014(7):694-708.

Chen J, Liao S, Xiao Z, et al. The development and improvement of immunodeficient mice and humanized immune system mouse models. *Front Immunol*. 2022;13:1007579.

3) The wording from the middle paragraph on pg 9 is not correct "To further ... receptor independent" is wrong. The authors switched to MKN45 cells and as such were not inducing EMT anymore with Ephrin A1 O/E as they did with NCI-N87 cells, they are suppressing an innate tendency for EMT in MKN45 cells with Ephrin A1 knockdown. Induction of EMT needs to have been done with O/E of Ephrin A1 in NCI-N87 cells.

Thanks a lot for pointing out this important concern. We fully agreed with your suggestion and have corrected our expression in page 9 of our revised manuscript. In addition, we tried to detect whether the EMT induction by Ephrin A1 in NCI-N87 and AGS cells are independent of EphA2. Unfortunately, NCI-N87 cells is tightly knit epithelial cells with very slow growing and we did not get *EphA2*-KO NCI-N87 cells successfully. However, we obtained *EphA2*-KO cells in AGS cells. Western blot and DNA sequencing data showed that the AGS-*EphA2*-KO cells were established successfully (**Figs. R15A-C**). Then we ectopically expressed Ephrin A1 and found that Ephrin A1 significantly promoted EMT process in AGS-*EphA2*-KO cells (**Fig. R15D**), indicating that Ephrin A1 inducing EMT of gastric cancer cells was EphA2 receptor independent. We have included **Fig. R15** as **new Fig. EV4** in our revised manuscript, and the corresponding description was in pages 9-10.

Figure for reviewers removed

Moreover, to further explore whether the EMT induction in NCI-N87 cells is EphA2 independent, we knocked down EphA2 by siRNA in NCI-N87 cells and detected the function of Ephrin A1 in inducing EMT. The results showed that overexpression of Ephrin A1 significantly induced EMT in EphA2-knockdown cells (**Fig. R16**). These data suggested that the function of Ephrin A1 in inducing EMT of NCI-N87 cells was not dependent on EphA2 receptor.

Figure for reviewers removed

4) Why did Ephrin A1 expression dramatically decreased with knockout of EphA2 (Fig. 2C). Isn't this going to be partially responsible for the phenotype? If EphA2 is knocked out in NCI-N87 cells are they resistant to EMT induction?

Thanks for your valuable concern. It is really an interesting phenomenon. EphA2 is a primary receptor of Ephrin A1 and EphA2-Ephrin A1 signaling is complex and dependent on cell type and microenvironment (Beauchamp et al., 2012; Wykosky et al., 2008). Here, we found that although Ephrin A1 inducing EMT is independent of EphA2 in gastric cancer cells, the expression level of Ephrin A1 was changed in *EphA2* KO cells, indicating other potential feedback regulation mechanisms between Ephrin A1 and EphA2 at the expression level, which really worth further studies.

In addition, according to the reviewer's good suggestion, we tried to generate *EphA2* KO cells in NCI-N87 cell. Unfortunately, NCI-N87 cells is tightly knit epithelial cells with very slow growing (ATCC, <https://www.atcc.org/products/crl-5822>) and we did not get *EphA2*-KO NCI-N87 cells successfully. Therefore, we performed EphA2 knockdown by siRNA and detected the EMT induction by Ephrin A1. The results showed that Ephrin A1 induced EMT in EphA2-knockdown cells (**Fig. R17**), suggesting the function of Ephrin A1 in inducing EMT of NCI-N87 cells was not dependent on EphA2 receptor.

Figure for reviewers removed

References:

Beauchamp A, Debinski W. Ephs and ephrins in cancer: ephrin-A1 signalling.

Semin Cell Dev Biol. 2012;23(1):109-115.

Wykosky J, Debinski W. The EphA2 receptor and ephrinA1 ligand in solid tumors: function and therapeutic targeting. *Mol Cancer Res.* 2008;6(12):1795-1806.

5) EphA2 has been shown to interact with EGFR [Kim (2022) *Cell death and disease* 13:528]. As such the authors should perform the Ephrin A1 pulldown in the MKN45 EphA2 knockout cells to demonstrate that they are not simply pulling down the ternary complex of Ephrin A1-EphA2-EGFR.

Thanks very much for your constructive suggestion. To address your question, we performed immunoprecipitation assay in MKN45-EphA2-KO cells and found that Ephrin A1 interacted with EGFR in the absent of EphA2 (**Figs. R18A-B**). In our current manuscript, we have added **Fig. R18B** as **new Fig.3C**, and the corresponding description was included in Pages 10.

Figure for reviewers removed

6) Why did the authors switch to MKN45 cells in Figure 2? The NCI-N87 EMT experiments seemed much more dramatic.

Thank the reviewer for the valuable concern. In our study, NCI-N87 cells have very low expression level of Ephrin A1 and were used for Ephrin A1 exogenous expression assays. Whereas MKN45 cells have high expression level of Ephrin A1 and used for Ephrin A1 knock-down experiments (Fig 1 and Fig EV1).

According to the reviewer's good suggestion, we further detected the receptors binding to Ephrin A1 in NCI-N87 cells and found that Ephrin A1 interacted with EphA2 (**Fig. R19A**) and EGFR (**Fig. R19B**), which was consistent with the results in MKN45 cells. We have included **Fig. R19** in our revised manuscript as **new Fig. 2A and Fig. 3B**, and the corresponding description was also included in pages 9 and 10 of our current manuscript.

Figure for reviewers removed

7) Did the EGFR mutants localize to the apical membrane like wild-type? Cellular mislocalization might explain the lack of binding. Anti-Myc IF on these cells should be performed.

This is really a good suggestion. it is important to examine the localization of EGFR mutants to determine whether the loss of binding is due to the absence of specific domains. According to your kind suggestion, we expressed EGFR and truncated EGFRs in cells and detected the localization of these proteins by immunofluorescence with anti-Myc antibody. The results showed that truncated EGFR (Δ ECD, Δ D1, Δ D3, Δ D1+D3) proteins distributed in both cell membrane and cytoplasm, which had similar localization pattern with full-length EGFR proteins in cells (**Fig. R20**).

Figure for reviewers removed

8) Why did the authors then switch to AGS cell lines in figure 3? The authors should repeat all experiments in Figure 2 with the AGS cells to argue that the phosphorylation changes are due to Ephrin A1-EGFR. What EphA2, A3, A4 are in AGS cells. Do these bind Ephrin? What do experiments with knockout of EphA2 (and A3, A4 if necessary) look like in AGS cells.

Thank you for your valuable suggestion. In our study, NCI-N87 and AGS cells have low expression levels of Ephrin A1 and were used for Ephrin A1 exogenous expression assays. Whereas MKN45 cells have high expression level of Ephrin A1 and used for Ephrin A1 knock-down experiments (Fig. EV1A). To proving EphrinA1 could activate EGFR and its pathway, we required to exogenous express Ephrin A1 or treatment with Ephrin A1 protein in cells. Therefore, AGS cells with low Ephrin A1 expression level and high transfection efficiency were used. To reconcile the cell lines used in the study, we rearranged main data from NCI-N87 and MKN45 cells in Figures and data from AGS cells in EV Figures in our revised manuscript.

According to the reviewer's suggestion, we detected the receptors binding to Ephrin A1 in AGS cells and found that Ephrin A1 interacted with EphA2 and EGFR (**Fig. R21A**), but not EphA1 and EphA4, which was consistent with the results in Fig2. Moreover, we generated *EphA2*-KO AGS cells by CRISPR-Cas9 system (**Fig. R21B**). Western blot and DNA sequencing data showed that the KO cells were established successfully (**Figs. R21C-D**). Then we ectopically expressed Ephrin A1 and found that Ephrin A1 significantly promoted EMT process both in wild type AGS cells and AGS-*EphA2*-KO cells (**Fig. R21E**). In addition, transwell migration and invasion assays showed that ectopic expression of Ephrin A1 markedly facilitated migration (**Figs. R21F-G**)

and invasion (**Figs. R21H-I**) abilities of AGS-*EphA2*-KO cells, which were consistent with the results in AGS cells. These data demonstrated that Ephrin A1 could promote EMT and metastasis abilities of AGS-*EphA2*-KO cells.

Considering the significance of these results, we have included **Fig. R21** as **new Fig. EV4** in our revised manuscript, and the corresponding description was in pages 9-10.

Figure for reviewers removed

Furthermore, we evaluated whether Ephrin A1 could activate EGFR signaling. The results showed that ectopic expression of Ephrin A1 significantly increased the phosphorylation level of EGFR (**Fig. R22A**) and promoted the activation of EGFR downstream signaling (**Fig. R22B**) both in AGS and AGS-*EphA2*-KO cells. Together, these results indicated that Ephrin A1 interacted with EGFR and induced EMT and metastasis of gastric cancer cells through activation of EGFR signaling, which was EphA2 receptor independent. Here, we have added **Fig. R22** as **new Figs. EV5H-I** in our revised manuscript, and the corresponding description was included in Page 12.

Figure for reviewers removed

9) Which cell line is in figure 3H? - Figure legend says gastric cancer cells and the authors keep on switching between three different lines so it is unclear what cell line this is.

We apologize for not making this clear in the manuscript and thank you for highlighting this point. In Fig 3H, we observed the increasing internalization of

EGFR in the cytoplasm after Ephrin A1-Fc protein treatment in MKN45 cells. Furthermore, we performed this experiment in AGS cells, and also found that Ephrin A1-Fc protein treatment significantly promoted the internalization of EGFR (**Fig. R23**).

Figure for reviewers removed

10) Does EGFR get phosphorylated and dimerize in MKN45 cells? This is important because the authors switched back to MKN45 cells at the end of figure 3 (I and J) and said there is downstream signaling from EGFR. But only showed phosphorylation and dimerization in AGS cells in response to Ephrin A1.

We really appreciate your insightful comments. To address your question, we conducted EGFR phosphorylation and dimerization assays in MKN45 cells. The results showed that knockdown of Ephrin A1 significantly inhibited the phosphorylation level (**Fig. R24A**) and the dimerization level of EGFR (**Fig. R24B**) in MKN45 cells. According to the reviewer's good suggestion, we have added **Fig. R24** as **new Figs. 4F-G** in our current manuscript, and the corresponding description was included in Page 12.

Figure for reviewers removed

11) Figure 4. The authors demonstrate that EMT is blocked with the drugs in AGS cells, but then test the soft agar and TEM in NCI-N87 cells. Please demonstrate that EMT is blocked in the NCI-N87 cells with these drugs when Ephrin is added.

Thanks for your constructive suggestion. We treated Ephrin A1-overexpressing NCI-N87 cells with inhibitors U0126 (ERK inhibitor) or defactinib (FAK inhibitor), and then detected the expression of EMT markers. Western blot assay showed that the EMT process induced by Ephrin A1 was suppressed in NCI-N87 cells treated with either U0126 or defactinib (**Fig. R25**), suggesting that Ephrin A1 may activates EMT of NCI-N87 cells through activation of EGFR downstream pathway. In our current manuscript, we have added **Fig. R25** as **new Fig. 5C**, and the corresponding description was included in Page 13.

Figure for reviewers removed

12) Figure 5. Quantitation of the western blots (throughout) would be useful. Looks like ZEB1 and N-Cad decrease in the EGFR KO (panel B). Does EGFR knockout in NCI-N87 cells block EMT with Ephrin A1.

Thank you for the valuable suggestion. We performed protein quantitation of all WB results in our revised manuscript. After quantitation, we found that knockdown of Ephrin A1 did not inhibit N-cadherin in EGFR KO cells and the decrease proportion of ZEB1 in KO cells was apparently less than WT cells.

According to the reviewer's suggestion, we tried to generate *EGFR*-KO NCI-N87 cells. Unfortunately, because NCI-N87 cells is tightly knit epithelial cells with very slow growing (ATCC, <https://www.atcc.org/products/crl-5822>), we did not get *EGFR* knockout NCI-N87 cells successfully. Therefore, we performed EGFR knockdown by siRNA and detected the EMT induction by Ephrin A1 cells. The results showed that Ephrin A1 failed to induce EMT in EGFR-knockdown cells (**Fig. R26**), suggesting that Ephrin A1 induced EMT of gastric cancer cells through EGFR receptor.

Figure for reviewers removed

13) Figure 6. The authors have not demonstrated that EGFR signaling (phosphorylation, dimerization, and downstream targets) is altered in NCI-N87 cells. These were shown in AGS and MKN45 cells.

Thanks for your constructive comments. As you suggested, we detected the expression levels of EGFR phosphorylation, dimerization and downstream targets in NCI-N87 cells. Our results showed that ectopic expression of Ephrin A1 significantly increased the phosphorylation and the dimerization levels of EGFR (**Figs. R27A-B**) and activated EGFR downstream signaling (**Fig. R27C**) in NCI-N87 cells. Moreover, the Ephrin A1 induced EGFR signaling were significantly suppressed after treatment with EGFR tyrosine kinase inhibitor (TKI) erlotinib (**Fig. R27D**). Here, we have added **Fig. R27** as **new Figs. 4C-E and Fig. 5H** in our revised manuscript, and the corresponding description was included in Pages 11 and 13.

Figure for reviewers removed

14) Can the authors demonstrate that inhibition of EphA2 does not block the Ephrin phenotype. Some inhibitors listed here: Xiao (2020) Journal of hematology and oncology 13:114. Dasatazinib: Chang et al (2008) Br J Cancer 99(7): 1074.

Thank you very much for your constructive comments. According to the provided literature, dasatinib was a widely used EphA2 inhibitor. Therefore, to determine whether the function of Ephrin A1 in gastric cancer cells was independent of EphA2, we used dasatinib to block EphA2 activation. We found that Ephrin A1 promoted the migration (**Figs. R28A, B**) and invasion (**Figs. R28C, D**) abilities of AGS cells treated with dasatinib.

Dasatinib is a multi-targeted kinase inhibitor mainly targeting Bcr-Abl, Src, c-kit, PDGFR, EphA2 and so on (Xiao et al., 2020). A variety of studies have demonstrated that dasatinib induced apoptosis and inhibited cell growth, migration and invasion of cancer cells (Chang et al., 2008; Buettner et al., 2008; Okabe et al., 2011). Therefore, the complex multi-target effects of dasatinib may be the reason why the induction effect of Ephrin A1 was not as obvious as DMSO group.

Figure for reviewers removed

Moreover, to further verify that the function of Ephrin A1 in inducing EMT and metastasis of gastric cancer cells is EphA2 independent, we generated *EphA2* knockout cell line in AGS cells (**Fig. R29A**). Western blot and DNA sequencing data showed that the KO cells were established successfully (**Figs. R29B, C**). Then we ectopically expressed Ephrin A1 and found that Ephrin A1 significantly promoted EMT process in AGS-*EphA2*-KO cells (**Fig. R29D**). In addition, the results of transwell migration and invasion assays showed that ectopic expression of Ephrin A1 markedly facilitated migration (**Figs. R29E, F**) and invasion (**Figs. R29G, H**) abilities of AGS-*EphA2*-KO cells, which were

consistent with the results in AGS cells. These data demonstrated that Ephrin A1 promoted EMT and metastasis abilities of AGS cells, which was independent of EphA2. Here, we have included **Fig. R29** as **new Fig. EV4** in our revised manuscript, and the corresponding description was in pages 9-10.

Figure for reviewers removed

References:

Xiao T, Xiao Y, Wang W, Tang YY, Xiao Z, Su M. Targeting EphA2 in cancer. *J Hematol Oncol*. 2020;13(1):114.

Buettner R, Mesa T, Vultur A, Lee F, Jove R. Inhibition of Src family kinases with dasatinib blocks migration and invasion of human melanoma cells. *Mol Cancer Res*. 2008;6(11):1766–1774.

Chang Q, Jorgensen C, Pawson T, Hedley DW. Effects of dasatinib on EphA2 receptor tyrosine kinase activity and downstream signaling in pancreatic cancer. *Br J Cancer*. 2008;99(7):1074-1082.

Okabe S, Tauchi T, Tanaka Y, Ohyashiki K. Dasatinib preferentially induces apoptosis by inhibiting Lyn kinase in nilotinib-resistant chronic myeloid leukemia cell line. *J Hematol Oncol*. 2011;4(1):32.

Minor:

-Re: Fig 3A. Ideally, the reverse IP would be done with EGFR (lacking the Fc region)

https://www.rndsystems.com/products/recombinant-human-egfr-protein_1095-er. It is not very expensive at 150 dollars (1095-ER)

We thank you very much for the valuable concern. As you suggested, we used Ephrin A1-Fc protein to pull down EGFR protein and confirmed that Ephrin A1 was able to bind to EGFR (**Fig. R30**). We have included **Fig. R30** in our revised manuscript as a **new Fig. 4B**, and the corresponding description was included in Page 11.

Figure for reviewers removed

-I do NOT doubt that knockouts were made based on the western blots presented; however, CRISPR makes random indels so it would be rare to have the single indel on both chromosome (or all if they are hyper diploid or tetraploid) as suggested by the figures.

Thank you for the insightful question. We fully agree with the reviewer's comment that it was rare to have the single indel on both chromosome by CRISPR-cas9 system, therefore we screened numerous monoclonal cell lines in our study. Except for identification of KO cells by western blots, we also extracted the genomic DNA and performed PCR and Sanger sequencing of the target sites. The sequencing data showed that MKN45-*EphA2*-KO cells were 1 bp insertion and MKN45-*EGFR*-KO cells were 5 bp deletions compared with wild type cells (**Fig. R31**), which causing frame shift mutation. These results demonstrated that the knock out cells were conducted successfully.

Figure for reviewers removed

Dear Dr Zhuo,

Thank you for submitting your revised manuscript (EMBOJ-2024-117725R) to The EMBO Journal, as well for your patience with our response. Your amended study was sent back to the three referees for their scientific re-evaluation, and we have received detailed comments from two of them, which I enclose below. Please note that while referee #2 was not able to re-evaluate your revised work at this time, we have assessed your response to this expert editorially and found it to be satisfactory. As you will see, the other experts state that the work has been substantially enhanced by the revisions and they are now broadly in favour of publication, pending minor revision.

Thus, we are pleased to inform you that your manuscript has been accepted in principle for publication in The EMBO Journal.

Please carefully consider the remaining minor points raised by reviewer #3 regarding statements made on results and wording of the manuscript and adjust the text where appropriate.

We also now need you to take care of a number of issues related to formatting and data presentation as detailed below, which should be addressed at re-submission.

Please contact me at any time if you have additional questions related to below points.

As you might have seen on our web page, every paper at the EMBO Journal now includes a 'Synopsis', displayed on the html and freely accessible to all readers. The synopsis includes a 'model' figure as well as 2-5 one-short-sentence bullet points that summarize the article. I would appreciate if you could provide this figure and the bullet points.

Thank you for giving us the chance to consider your manuscript for The EMBO Journal. I look forward to your final revision.

Again, please contact me at any time if you need any help or have further questions.

Kind regards,

Daniel Klimmeck

>> Authors: please revisit e-mail contact provided in our system for co-author Y.D. .

>> Author Contributions: Remove the author contributions information from the manuscript text. Note that CRediT has replaced the traditional author contributions section as of now because it offers a systematic machine-readable author contributions format that allows for more effective research assessment. and use the free text boxes beneath each contributing author's name to add specific details on the author's contribution.

More information is available in our guide to authors.
<https://www.embopress.org/page/journal/14602075/authorguide>

>> Adjust the title of the 'Conflict of Interest' section to 'Disclosure and Competing Interests Statement' and move after

Acknowledgements.

>> Correct order of manuscript sections: Abstract / Keywords / Introduction / Results / Discussion / Methods / Data Availability / Acknowledgements / Disclosure and competing interests statement // References / Figure legends / Tables and their legends / Expanded View Figure legends

>> Author Checklist: please complete all selections in the 'Experimental Animals' section.

>> Figures in separate files: the EV figures should also be uploaded as individual, high-resolution figure files.

>> Funding: merge the Funding information in the manuscript text with the Acknowledgements section.

>> Data availability section: change the statement to: 'No data amenable to large-scale repository deposition were generated in this study.' .

>> Add a Reagents and Tools table to the Methods section, as a separate file using the existing template in the Guide For Authors, listing key reagents, experimental models, software and relevant equipment.

>> Consider additional changes and comments from our production team as indicated below:

- Figure legends:

1. Please note that the exact p values are not provided in the legends of figures 1g, j, m, o, q; 2f, h, j, m; 5e, g, j, l; 6d-e; 7b; EV 1c-d, f; EV 2b, d, f, h, j, l; EV 3b, d, f, l; EV 4f, h; EV 6c, e, h, j.

2. Please indicate the statistical test used for data analysis in the legend of figure 7d.

3. Please note that in figures 2f, h, j, m; EV 3j; there is a mismatch between the annotated p values in the figure legend and the annotated p values in the figure file that should be corrected.

4. Please note that information related to n is missing in the legends of figures 2f, h, m; 6d-e; EV 3j, l.

5. Please note that axis gaps are not labeled appropriately in figures EV 1c, f.

Referee #1:

The authors have thoroughly responded to the points raised in the last round of review and significantly improved the quality of the manuscript for publication.

Referee #3:

Major concerns were addressed in the extensive revision.

I would recommend that the authors should specifically state that EphA2 KO results in a dramatic reduction in Ephrin A1 probably somewhere before the "Moreover" on page 10 (Immediately before or in the preceding paragraph). As the reduction in Ephrin A1 in the EphA2 KO is more potent than the siRNA against Ephrin A1. It is perfectly fine to say there is added benefit to knocking down with siRNA in the KO because there is, which there is.

Also could consider adding "although EphA2 KO dramatically decreases Ephrin A1 levels" after "independent of EphA2" in the

discussion on page 16. One might argue based on your data that this is a potential mechanism by which EphA2 functions to induce EMT in Fattet et al. 2020 and Huang et. al. 2014 - mentioned in the prior sentence, e.g. by increasing increasing Ephrin A1 levels and thus inducing EGFR signaling instead of EphA2 actually doing this on its own ... (something that wasn't measured in those studies).

Several instances of awkward sentences (the meaning could be ascertained but the wording needs some work) that can easily be fixed by a copyeditor.

Response to Reviewers' and editors' comments:**Reply to Referee #1:**

The authors have thoroughly responded to the points raised in the last round of review and significantly improved the quality of the manuscript for publication.

Thank you very much for reviewing our manuscript and providing valuable suggestions to help us improve our work.

Reply to Referee #3:

Major concerns were addressed in the extensive revision.

I would recommend that the authors should specifically state that EphA2 KO results in a dramatic reduction in Ephrin A1 probably somewhere before the "Moreover" on page 10 (Immediately before or in the preceding paragraph). As the reduction in Ephrin A1 in the EphA2 KO is more potent than the siRNA against Ephrin A1. It is perfectly fine to say there is added benefit to knocking down with siRNA in the KO because there is, which there is.

We really appreciate your insightful recommendations on strengthening our manuscript.

(1) According to your kind suggestions, we have added the description of *EphA2* KO results in a dramatic reduction in Ephrin A1 to the results section (page 9). The descriptions were as following.

Results (page 9): Interestingly, *EphA2* KO results in a dramatic reduction in Ephrin A1, which causing significant inhibition of EMT in *EphA2*-KO MKN45 cells compared with WT cells.

Also could consider adding "although EphA2 KO dramatically decreases Ephrin A1 levels" after "independent of EphA2" in the discussion on page 16. One might argue based on your data that this is a potential mechanism by which EphA2 functions to induce EMT in Fattet et al. 2020 and Huang et. al. 2014 – mentioned in the prior sentence, e.g. by increasing increasing Ephrin A1 levels and thus inducing EGFR signaling instead of EphA2 actually doing this on its own ...(something that wasn't measured in those studies).

(2) We have also added "although *EphA2* KO dramatically decreases Ephrin A1 levels" in the discussion section of our current manuscript (page 17). The descriptions were as following.

Discussion (page 17): In this study, we found the function of Ephrin A1 in inducing

EMT was independent of EphA2 although *EphA2* KO dramatically decreases Ephrin A1 levels.

Thanks a lot for your valuable concern. The study in Fattet et al. 2020 showed that high ECM stiffness promoted ligand-independent phosphorylation of EphA2, which activated LYN kinase to promote EMT and invasion of breast cancer. And in Huang et al. 2014, the study indicated that EphA2 promoted EMT of gastric cancer cells through activation of Wnt/ β -catenin signaling. However, in our study, we found that *EphA2* KO reduced the expression level of Ephrin A1. Thus, there may be another potential mechanism of EphA2 in regulating EMT through Ephrin A1/EGFR signaling, which is worthy of exploration.

Several instances of awkward sentences (the meaning could be ascertained but the wording needs some work) that can easily be fixed by a copyeditor.

(3) Thanks very much for your attention to the text, we have polished our manuscript carefully to make it easy to follow.

Dear Dr Zhuo,

Thank you for submitting the revised version of your manuscript. I have now evaluated your amended manuscript and concluded that the remaining minor concerns have been sufficiently addressed.

I am thus pleased to inform you that your manuscript has been accepted for publication in the EMBO Journal.

Related I would like to ask for your consent on keeping the referee figures included in this file.

On a different note, I would like to alert you that EMBO Press offers a format for a video-synopsis of work published with us, which essentially is a short, author-generated film explaining the core findings in hand drawings, and, as we believe, can be very useful to increase visibility of the work. Please see the following link for representative examples and their integration into the article web page:

<https://www.embopress.org/doi/full/10.15252/emj.2019103932>

Best regards,

Daniel Klimmeck

Daniel Klimmeck, PhD
Senior Editor
The EMBO Journal
EMBO
Postfach 1022-40
Meyerohofstrasse 1
D-69117 Heidelberg
contact@embojournal.org
Submit at: <http://emboj.msubmit.net>